# HyperCT: Low-Rank Hypernet for Unified Chest CT Analysis

**Fengbei Liu**[1]                                                     FL453@CORNELL.EDU
**Sunwoo Kwak**[1]                                                     SK3355@CORNELL.EDU
**Hao Phung**[1]                                                        HTP26@CORNELL.EDU
**Nusrat Binta Nizam**[1]                                               NN284@CORNELL.EDU
**Ilan Richter**[2]                                              IR2498@CUMC.COLUMBIA.EDU
**Nir Uriel**[2]                                                 NU2126@CUMC.COLUMBIA.EDU
**Hadar Averbuch-Elor**[1]                                         HADARELOR@CORNELL.EDU
**Deborah Estrin**[1,3]                                              DESTRIN@CORNELL.EDU
**Mert R. Sabuncu**[1,3]                                            MSABUNCU@CORNELL.EDU

[1] *Cornell Tech and Cornell University*
[2] *Columbia University Irving Medical Center*
[3] *Weill Cornell Medicine*

**Editors:** Accepted for publication at MIDL 2026

## Abstract

Non-contrast chest CTs offer a rich opportunity for both conventional pulmonary and op-portunistic extra-pulmonary screening. While Multi-Task Learning (MTL) can unify these diverse tasks, standard hard-parameter sharing approaches are often suboptimal for mod-eling distinct pathologies. We propose **HyperCT**, a framework that dynamically adapts a Vision Transformer backbone via a Hypernetwork. To ensure computational efficiency, we integrate Low-Rank Adaptation (LoRA), allowing the model to regress task-specific low-rank weight updates rather than full parameters. Validated on a large-scale dataset of radiological and cardiological tasks, HyperCT outperforms various strong baselines, of-fering a unified, parameter-efficient solution for holistic patient assessment. Our code is available at https://github.com/lfb-1/HyperCT.

**Keywords:** Chest CT, Hypernetwork, Low-Rank Adaptation, Multi-task Learning

## 1. Introduction

Non-contrast chest computed tomography (CT) is a fundamental modality of modern radi-ology, serving as the standard for pulmonary screening due to its high spatial resolution and rapid acquisition. This has led to the creation of vast archives of chest CT data. While these scans are primarily used for **conventional screening** tasks such as detecting pulmonary nodules or emphysema (Hamamci et al., 2024; Anouk Stein et al., 2018), they capture a rich anatomical context, including the heart, great vessels, and upper abdominal organs. This has given rise to an emerging paradigm of **opportunistic screening** (Pickhardt et al., 2023), where a single CT exam is repurposed to screen for extra-pulmonary conditions. In this work, we focus on cardiac structural and functional assessments typically derived from echocardiography—conditions not traditionally predictable from CT, yet the heart is fully included in the chest CT field of view. This represents a powerful shift toward holistic patient assessment, aiming to extract maximum clinical value from existing data.

Despite the potential for comprehensive health profiling, current chest CT screening approaches remain isolated, typically designed either entirely for conventional tasks or a single opportunistic target (Hamamci et al., 2024; Huang et al., 2025). A unified framework capable of performing both simultaneously remains a critical, unaddressed gap. To bridge this gap, we utilize Multi-task learning (MTL), where a single model jointly learns from both conventional and opportunistic labels. However, standard MTL pipelines still struggle to effectively mitigate task interference, which can result in performance degradation on certain tasks (Kendall et al., 2018; Navon et al., 2022; Lin et al., 2021). These methods assume tasks inherently *conflict*—that is, they compete and interfere with each other—and focus on mitigating negative transfer. We argue this assumption is misaligned with medical screening, where findings are often synergistic and comorbid (e.g., cardiac enlargement frequently co-occurs with pulmonary congestion). The core motivation of this paper is that the central challenge in opportunistic screening is not merely to balance competing tasks, but to design a model that can explicitly learn and leverage the synergistic relationships between diverse clinical domains and improve overall diagnostic performance.

Accordingly, we propose **HyperCT**, a novel framework that achieves unified screening by dynamically generating task-specific parameters. Our approach uses a Hypernetwork (Ha et al., 2016) that takes a task's identity as input and outputs the weights needed to adapt a base model for a specific target. This mechanism enables flexible task-adaptive parameter sharing, moving beyond the rigid backbones of standard MTL. To make this approach computationally feasible for high-capacity architectures like Vision Transformers (ViT) (Dosovitskiy, 2020), we integrate Low-Rank Adaptation (LoRA) (Hu et al., 2022) into the hypernetwork design. Instead of generating full-rank weight matrices, our method regresses low-rank updates, dramatically reducing the complexity of the hypernetwork while preserving the expressive power needed for a diverse set of screening tasks.

We demonstrate the effectiveness of our proposed HyperCT framework on a large-scale curated dataset comprising both conventional and opportunistic screening tasks derived from non-contrast chest CTs. Our results show that the model outperforms standard MTL baselines while achieving comparable performance to dedicated single-task models. This eliminates the need to train separate models for each task while maintaining constant parameter count, highlighting its potential as a unified solution for comprehensive chest CT screening. Our contributions can be summarized as follows:

- We present the first unified framework for joint conventional and opportunistic chest CT screening, bridging 18 pulmonary and 7 cardiovascular tasks that have previously been addressed in isolation.

- We integrate LoRA into the hypernetwork design, enabling efficient generation of task-specific weights for high-capacity architectures like ViTs—overcoming the scalability limitations that have restricted prior hypernetwork applications to small architectures or simple adapters.

- We provide comprehensive validation across retrospective, prospective, and multi-institutional cohorts, demonstrating that HyperCT outperforms MTL baselines while matching single-task model performance.

## 2. Related Works

**Chest CT screening.** The clinical utility of non-contrast chest CT was established by the National Lung Screening Trial (NLST) (Team, 2011), which demonstrated a significant mortality benefit in lung cancer screening. This study catalyzed the application of deep learning to automate radiological interpretation, initially focusing on pulmonary nodules (Setio et al., 2017) and expanding to diffuse chronic diseases like emphysema (Humphries et al., 2020; Li et al., 2023). Recently, the field has recognized the rich, extra-pulmonary information available in these scans, leading to the paradigm of opportunistic screening for conditions such as esophageal cancer (Yao et al., 2022) and cardiovascular risk (Raikhelkar et al., 2025). However, these two powerful screening paradigms have evolved largely in parallel. Current models are typically developed in isolation, focusing either on a suite of conventional findings or a single opportunistic target. A unified framework capable of performing both simultaneously remains a critical, unaddressed gap.

**Multi-task learning.** Multi-task learning (MTL) aims to improve performance by jointly learning multiple related tasks (Caruana, 1997). **Optimization-based** approaches focus on balancing task learning through loss weighting—such as Uncertainty Weighting (UW) (Kendall et al., 2018), Random Loss Weighting (RLW) (Lin et al., 2021), and MGDA (Sener and Koltun, 2018), or gradient manipulation strategies like GradNorm (Chen et al., 2018) and Nash-MTL (Navon et al., 2022). These techniques assume tasks are competing and aim to mitigate negative interference, which is a perspective misaligned with medical screening where findings are often synergistic and comorbid. **Architecture-based** approaches, including hard/soft parameter sharing (Misra et al., 2016; Ruder et al., 2019), mixture-of-experts (Chen et al., 2023), and neural architecture search (Guo et al., 2020), offer alternatives but often rely on heuristic designs tailored for CNNs, making adaptation to modern Vision Transformers non-trivial.

**Hypernetworks.** Hypernetworks (Ha et al., 2016) are a class of neural architectures designed to generate the weights of a "base" model. Recently, this approach has gained traction in Multi-Task Learning (MTL) through the use of task-conditioned hypernetworks. Mahabadi et al. (Mahabadi et al., 2021) demonstrated that hypernetworks can facilitate knowledge sharing across tasks while generating task-specific adapter layers, achieving state-of-the-art results in NLP benchmarks. Similarly, Navon et al. (Navon et al., 2020) utilized hypernetworks to approximate the Pareto front, effectively addressing gradient conflicts in diverse multi-objective settings ranging from fairness constraints to image segmentation. In medical imaging, related conditioning mechanisms have been explored: FiLM (Perez et al., 2018) introduces feature-wise affine transformations for visual reasoning, MAC-ReconNet (Ramanarayanan et al., 2020) applies hypernetworks to multi-coil MRI reconstruction, MetaInv-Net (Zhao et al., 2020) uses meta-learning for inverse problems, and Hyper-GAN (Hoopes et al., 2021) leverages hypernetworks for deformable registration. The primary bottleneck for scaling hypernetworks is that their output size is tied to the target model's parameter count. This often makes the hypernetwork itself too large, limiting its application to small architectures or simple adapters and creating a major challenge for adapting large models like Vision Transformers (ViTs).

## 3. Method

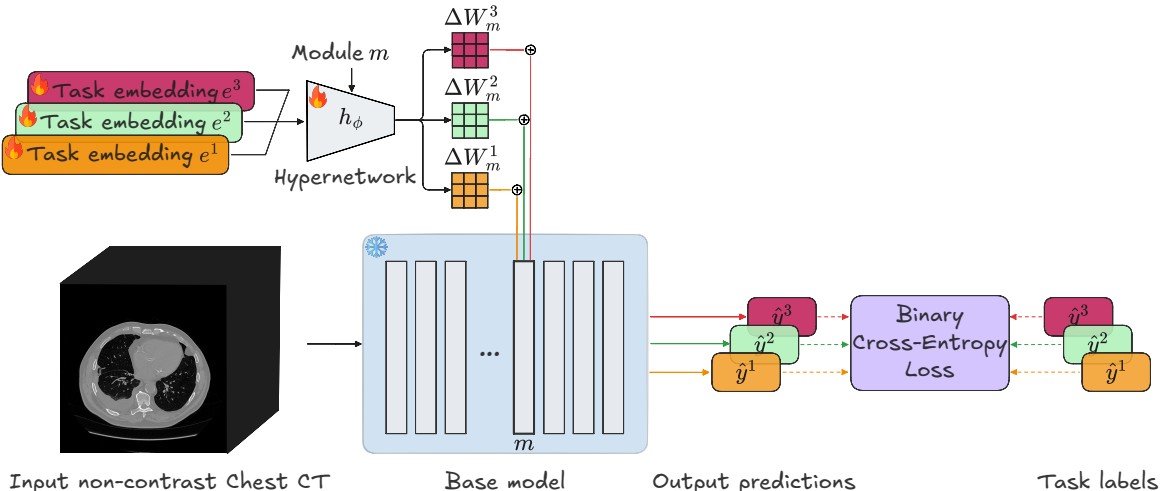

Figure 1: Overview of HyperCT. Given a set of learnable task embeddings, e.g., $\{\mathbf{e}^1, \mathbf{e}^2, \mathbf{e}^3\}$, a hypernet $h$ produces task-specific weight adjustments $\Delta\mathbf{W}^1, \Delta\mathbf{W}^2, \Delta\mathbf{W}^3$, which modulate the weights of the base model. The base model, produces task-specific predictions $\{\hat{\mathbf{y}}^1, \hat{\mathbf{y}}^2, \hat{\mathbf{y}}^3\}$. These outputs are compared with ground-truth task labels $\{\mathbf{y}^1, \mathbf{y}^2, \mathbf{y}^3\}$ via Binary Cross-Entropy Loss.

An overview of our proposed method is presented in Figure 1. The architectural framework includes a pre-trained backbone $f_\theta$ parameterized by $\theta = \{\mathbf{W}_1, \mathbf{W}_2, ..., \mathbf{W}_M\}$, in which $M$ denotes the number of total modules within the base model, and a Hypernetwork $h_\phi$ parameterized by $\phi$, which generates task-specific parameters for the base model. We denote the learnable task embeddings as $E = \{\mathbf{e}^1, \mathbf{e}^2, ..., \mathbf{e}^K\}$, where each $\mathbf{e}^k \in \mathbb{R}^{d_e}$ corresponds to a specific task representation which is processed by hypernet $h_\phi$ to generate task-conditioned parameters. Given an input CT scan $x \in \mathbb{R}^{H \times W \times Z}$ and a desired task $k$, where $Z$ represents the number of slices, and $H$ and $W$ denotes the spatial dimensions, our goal is to leverage those generated task-specific parameters to predict a binary label $\hat{y}^k \in \{0, 1\}$.

**Hypernetwork-based Weight Generation:** Unlike static multi-task learning, where a set of back-bone parameters $\theta$ is shared across tasks, we employ the Hypernetwork $h_\phi$ to dynamically regress the parameter of the base model $f_\theta$ conditioned on the task representation $\mathbf{e}^k$. Our objective is to generate a task-specific parameter set $\theta^k$ for each task $k$ as the following:

$$\theta^k = h_\phi(\mathbf{e}^k), \quad \hat{\mathbf{y}}^k = f_{\theta^k}(\mathbf{x}), \quad \mathcal{L} = \frac{1}{K} \sum_{k=1}^{K} \mathrm{BCE}(\hat{\mathbf{y}}^k, \mathbf{y}^k) \tag{1}$$

where $\mathcal{L}$ is the Binary Cross-Entropy (BCE) loss between a task-specific model prediction $\hat{y}^k$ and the ground truth $y^k$. This formulation allows the model to adaptively distribute the capacity based on the specific screening task encoded in $\mathbf{e}^k$. Here we assume each task is equally weighted. However, advanced weighting techniques can be seamlessly incorporated.

Instead of outputting all the base model weights together, in our implementation, the hypernetwork outputs the weights of each module, using a module indicator as an additional input. The module indicator is a learned vector-valued function $\phi_{\text{pos}}$ of the module index. Intuitively, $\phi_{\text{pos}}$ serves as a location indicator that tells the hypernetwork where in the ViT architecture to apply the generated weights. Without $\phi_{\text{pos}}$, the hypernetwork would receive only the task embedding and generate identical LoRA weights for all layers—this would fail, as different layers require different adaptations. By concatenating $\phi_{\text{pos}}(m)$ with the task embedding, the hypernetwork can generate layer-specific LoRA weights across all $M$ target modules. The hypernetwork $h_\phi$ generates weights for the target modules by iterating through $M$ target modules, using the task encoding vector $\mathbf{e}^k$ and module indicator $\phi_{\text{pos}}(m)$.

**Low-Rank Adaptation with Hypernetworks:** A practical implementation of the above framework requires careful consideration of the parameter efficiency of $h_\phi$. Directly regressing high-dimensional weight matrices $\mathbf{W}_m^k$ can lead to an explosion in the number of parameters within $h_\phi$, especially for recent ViT based architectures with large hidden dimensions. However, previous approaches naively consider only a few low-dimensional target modules can result in a significant loss of information (Navon et al., 2020), as the hypernetwork may not capture the full complexity of the task-specific weight distributions.

To mitigate this, we integrate Low-Rank Adaptation (LoRA) (Hu et al., 2022) as $h_\phi$ target modules. LoRA implements model adaptation via a low-dimensional intrinsic subspace. By decomposing the task-specific weight update into two low-rank matrices generated by the hypernetwork, we significantly constrain $h_\phi$'s output complexity while preserving the generalization ability of the overall pre-trained model. We perform a detailed analysis of the parameter efficiency in Appendix. Sec. E.

Specifically, for each target module weight $\mathbf{W}_m \in \mathbb{R}^{d_{in} \times d_{out}}$ with input dimension $d_{in}$ and output dimension $d_{out}$, we decompose it into a sum of a frozen pre-trained weight $\mathbf{W}_m^{\text{base}}$ and a low-rank update $\Delta\mathbf{W}_m^k$ generated by the hypernetwork. The overall forward pass and parameter generation are formulated as follows:

$$\mathbf{B}_m^k = h_\phi^B(\mathbf{e}^k, \phi_{\text{pos}}(m)) \quad \mathbf{A}_m^k = h_\phi^A(\mathbf{e}^k, \phi_{\text{pos}}(m))$$
$$\Delta\mathbf{W}_m^k = \mathbf{B}_m^k \mathbf{A}_m^k \quad \forall m \in \{1, \ldots, M\},$$
$$\theta^k = \left\{ \mathbf{W}_m^{\text{base}} + \frac{\alpha}{r} \Delta\mathbf{W}_m^k \right\}_{m=1}^M$$

where $\Delta\mathbf{W}_m^k$ is a low-rank matrix formed by two matrices $\mathbf{B}_m^k \in \mathbb{R}^{d_{in} \times r}$ and $\mathbf{A}_m^k \in \mathbb{R}^{r \times d_{out}}$ (with rank $r \ll \min(d_{in}, d_{out})$), each output by the hypernetwork $h_\phi$. This update weight is scaled by $\frac{\alpha}{r}$, where $\alpha$ is a predefined constant.

## 4. Experiments

### 4.1. Datasets curation

**Dataset Statistics.** We curated a large-scale dataset comprising 36,286 non-contrast chest CT scans collected from two major medical centers, Columbia University (CU) Medical Center and Weill Cornell Medical Center (WCM). The dataset is stratified into retrospective and prospective cohorts to rigorously evaluate the generalizability of HyperCT

across different clinical settings and time periods. The primary retrospective cohort consists of 34,058 scans acquired between 2011 and 2022. To assess cross-institutional robustness, we trained our models exclusively on the data from CU. Specifically, the 25,948 retrospective CU scans were partitioned into 18,213/2,561/5,174 training/validation/testing samples using a 70/10/20 split, with strict patient-level separation to prevent data leakage. The 8,110 retrospective scans from WCM were reserved strictly as an external test set. Additionally, we collected a prospective cohort of 2,228 scans acquired from 2023 to 2024 to serve as a temporal validation set, containing 1,411/817 exams from CU and WCM respectively.

**Task Definition and Labeling.** We defined a comprehensive set of $K = 25$ binary classification targets to evaluate on both conventional (18) and opportunistic (7) screening tasks. For the conventional tasks, we employed Llama3.1 (Grattafiori et al., 2024) to parse free-text radiology reports and extract binary pathology labels, which has been shown to outperform rule-based extractors (Dorfner et al., 2024; Kheradmand et al., 2025) (Prompt shown in Appendix. Sec. M). For the opportunistic tasks, we matched CT scans to corresponding echocardiography exams using patient identifiers within a maximum temporal window of $\pm180$ days; when multiple echocardiography exams were available, we selected the closest in time. We then defined binary ground truth labels for 7 clinically relevant measurements based on established thresholds determined in consultation with expert clinicians (Thresholds are shown in Appendix. Sec. L). The detail label statistics is shown in Appendix Sec. B. It is important to note that radiology report were not collected for the prospective cohort, and therefore the prospective evaluation focuses exclusively on the cardiology tasks.

## 4.2. Implementation Details

We implement our framework using Pytorch (Paszke et al., 2019). For data processing, each chest CT volume is resized to $H = W = 144$ and $Z = 165$ respectively. For the base model $f_\theta$, we adopt DINOv3 (Siméoni et al., 2025) as the pretrained frozen backbone architecture. We select ViT-base (Dosovitskiy, 2020) variant with 12 transformer layers, $D = 768$ hidden dimension. The hypernetwork $h_\phi$ is designed as a 3-layer MLP with hidden dimension $d_h = 64$ and $\phi_{\text{pos}}$ is an Embedding layer with $d_p = 64$. The task embeddings $\mathbf{e}^k$ are learnable vectors of dimension 512, initialized randomly. We set the LoRA rank $r = 16$ and scaling factor $\alpha = 16$ for all target modules to match the total trainable parameter size of baselines. We follow previous approaches and compress three consecutive slices as one 2D input to the base model (Gu et al., 2025; Lee et al., 2025).

During training, we use AdamW optimizer (Loshchilov and Hutter, 2017) with an initial learning rate of $1e^{-5}$ and weight decay of 0. We train the model for 20 epochs with a batch size of 8 on 1 NVIDIA A100 GPUs. The learning rate is decayed by a factor of 0.1 every 15 epochs. For each batch, we randomly sample one available task for each sample and compute BCE loss for the corresponding task prediction. We ablate this sampling strategy against inverse-prevalence weighted sampling in Appendix Sec. F, finding minimal performance difference ($<0.5\%$ AUC). During inference, we evaluate all available tasks for each sample and compute the Area Under Curve (AUC) for each task. Best model is selected by validation AUC.

For the MTL baseline implementation, we use the same base model and training hyperparameters for a fair comparison. We use LibMTL (Lin and Zhang, 2023), a publicly-available library, to implement various MTL baselines including Equal Weighting (EW), Uncertainty Weighting (UW) (Kendall et al., 2018), Random Loss Weighting (RLW) (Lin et al., 2021), Dynamic Weight Averaging (DWA) (Liu et al., 2019), and Multi-gradient Descent Algorithm (MGDA) (Sener and Koltun, 2018). Note we did not include recent gradient-based methods such as PCGrad (Yu et al., 2020) and Nash-MTL (Navon et al., 2022). These methods have $O(K^2)$ complexity per iteration due to pairwise gradient computations; with $K{=}25$ tasks, this becomes prohibitively slow for ViT-scale models on large datasets. Additionally, we compare with single-task learning baselines (STL) that separately finetune the base model for each task, using identical hyperparameters to HyperCT for fair comparison.

## 4.3. Retrospective Evaluation

Table 1 benchmarks HyperCT against six multi-task learning strategies, including gradient-balancing algorithms like MGDA and GLS in retrospective data. Across 25 tasks, HyperCT consistently achieves the highest performance, with an overall average AUC of 78.1% (CU) and 76.5% (WCM), surpassing the competitive MGDA baseline. We note that STL baselines show strong performance in retrospective data, but do not generalize as well in prospective evaluation (see below) and are significantly resource intensive (with each STL model containing roughly the same number of learnable models as the full HyperCT).

Table 1: Comprehensive comparison of AUC scores (%) on retrospective study. Best results among MTL methods are **bolded**, second best are underlined.

|  | Task | CU Test | | | | | | | | WCM Test | | | | | | | |
|---|---|---|---|---|---|---|---|---|---|---|---|---|---|---|---|---|---|
|  |  | STL | EW | UW | RLW | DWA | GLS | MGDA | HyperCT | STL | EW | UW | RLW | DWA | GLS | MGDA | HyperCT |
|  | *Overall Average* | 79.3 | 74.3 | 74.4 | 74.0 | 74.3 | 71.6 | 76.0 | **78.1** | 77.6 | 72.9 | 72.9 | 72.8 | 72.9 | 69.5 | 74.9 | **76.5** |
| Conventional | Med. Mat. | 88.2 | 83.6 | 83.2 | 81.8 | 83.6 | 76.4 | **85.9** | 85.8 | 89.3 | 85.5 | 85.2 | 84.4 | 85.5 | 78.9 | **87.6** | 87.3 |
|  | Art. Calc. | 82.0 | 79.5 | 79.4 | 79.0 | 79.5 | 76.8 | 80.5 | **81.9** | 75.3 | 73.9 | 73.9 | 73.2 | 73.9 | 71.2 | 74.4 | **76.0** |
|  | Cardiomeg. | 86.9 | 84.1 | 84.1 | 83.4 | 84.1 | 81.5 | 85.3 | **87.0** | 87.1 | 85.0 | 85.0 | 84.3 | 85.0 | 82.8 | 86.0 | **87.0** |
|  | Peri. Eff. | 70.2 | 68.0 | 68.0 | 67.5 | 68.0 | 63.9 | **69.4** | 68.5 | 73.3 | 67.4 | 67.4 | 67.3 | 67.4 | 62.6 | 69.9 | **71.1** |
|  | Cor. Art. Calc. | 90.1 | 83.8 | 83.8 | 83.3 | 83.8 | 81.6 | 84.8 | **88.2** | 84.6 | 78.7 | 78.6 | 77.8 | 78.6 | 76.1 | 79.3 | **82.3** |
|  | Hiatal Hernia | 70.8 | 59.6 | 60.1 | 58.5 | 59.6 | 59.2 | 61.3 | **67.6** | 70.6 | 56.7 | 57.3 | 55.6 | 56.7 | 56.0 | 60.5 | **68.8** |
|  | Lymphadenop. | 72.6 | 65.4 | 65.1 | 65.1 | 65.4 | 62.4 | **68.3** | 67.1 | 74.5 | 67.0 | 66.8 | 67.0 | 67.0 | 64.3 | **70.1** | 69.4 |
|  | Emphysema | 82.8 | 76.6 | 76.8 | 74.3 | 76.6 | 73.6 | 77.6 | **79.1** | 79.4 | 73.4 | 73.6 | 72.1 | 73.4 | 70.4 | 74.6 | **74.9** |
|  | Atelectasis | 80.3 | 74.8 | 74.6 | 74.0 | 74.8 | 72.1 | 76.7 | **77.7** | 80.5 | 76.3 | 76.2 | 76.1 | 76.4 | 74.0 | 78.0 | **78.2** |
|  | Lung Nodule | 68.1 | 68.8 | 68.6 | 68.5 | 68.8 | 66.3 | **70.0** | **70.0** | 62.3 | 65.0 | 64.8 | 64.8 | 65.0 | 62.3 | **65.6** | 64.4 |
|  | Lung Opacity | 78.8 | 75.6 | 75.4 | 74.7 | 75.6 | 73.3 | **78.2** | **78.2** | 77.5 | 75.1 | 74.9 | 74.4 | 75.1 | 72.5 | 77.4 | **78.0** |
|  | Pulm. Fibrosis | 85.9 | 84.2 | 84.3 | 83.5 | 84.2 | 83.4 | 84.7 | **85.2** | 81.8 | 80.3 | 80.3 | 79.8 | 80.2 | 79.6 | **81.0** | 80.6 |
|  | Pleural Eff. | 96.1 | 94.4 | 94.5 | 94.1 | 94.4 | 94.2 | 94.5 | **95.6** | 96.7 | 94.6 | 94.7 | 94.3 | 94.6 | 94.5 | 94.7 | **95.9** |
|  | Mosaic Attn. | 70.2 | 61.2 | 61.6 | 60.4 | 61.3 | 56.8 | 65.7 | **71.6** | 67.7 | 60.3 | 60.7 | 59.5 | 60.4 | 57.3 | 64.4 | **68.1** |
|  | Peribronchial | 66.9 | 62.7 | 62.5 | 62.1 | 62.7 | 59.5 | 64.9 | **66.3** | 65.5 | 63.7 | 63.4 | 62.8 | 63.7 | 60.3 | 66.3 | **66.5** |
|  | Consolidation | 87.9 | 83.2 | 83.1 | 82.3 | 83.2 | 80.3 | 85.0 | **86.3** | 83.9 | 78.9 | 78.9 | 78.0 | 78.9 | 75.7 | 80.5 | **82.0** |
|  | Bronchiectasis | 81.7 | 78.6 | 78.7 | 78.5 | 78.6 | 77.1 | 79.8 | **80.6** | 78.2 | 74.6 | 74.8 | 74.2 | 74.6 | 73.1 | 75.8 | **76.8** |
|  | Septal Thick. | 76.0 | 74.9 | 74.9 | 73.9 | 74.9 | 72.4 | **76.0** | 75.8 | 78.0 | 77.8 | 77.7 | 77.5 | 77.8 | 74.1 | **79.1** | 79.3 |
|  | *Group Avg.* | 79.8 | 75.5 | 75.5 | 74.7 | 75.5 | 72.8 | 77.1 | **78.5** | 78.1 | 74.1 | 74.1 | 73.5 | 74.1 | 71.4 | 75.8 | **77.0** |
| Opportunistic | Red. RV | 79.9 | 74.1 | 74.2 | 73.3 | 74.1 | 70.3 | 75.3 | **77.5** | 80.1 | 74.8 | 74.8 | 74.4 | 74.8 | 71.4 | 76.4 | **77.9** |
|  | Red. LV | 80.0 | 70.7 | 70.8 | 70.9 | 70.7 | 67.3 | 73.2 | **77.0** | 77.9 | 70.5 | 70.5 | 70.5 | 70.5 | 66.2 | 72.8 | **74.6** |
|  | Pulm. HTN | 71.2 | 71.6 | 71.4 | 70.7 | 71.6 | 68.5 | **72.9** | 72.7 | 71.6 | 65.0 | 69.8 | 70.0 | 70.1 | 67.6 | 71.4 | **72.0** |
|  | Atrial Enlg. | 83.9 | 75.8 | 75.9 | 75.8 | 75.8 | 69.8 | 78.2 | **83.0** | 82.3 | 74.1 | 74.2 | 74.3 | 74.1 | 68.6 | 76.6 | **79.9** |
|  | Vent. Enlg. | 83.8 | 68.6 | 69.2 | 69.9 | 68.6 | 62.7 | 72.0 | **80.4** | 75.0 | 61.0 | 61.4 | 61.6 | 61.0 | 58.5 | 64.5 | **73.1** |
|  | LA Pressure | 75.6 | 74.2 | 74.0 | 73.3 | 74.2 | 70.6 | 76.0 | **77.1** | 75.6 | 73.7 | 73.5 | 73.3 | 73.7 | 70.6 | 75.8 | **77.1** |
|  | RA Pressure | 72.1 | 63.8 | 64.6 | 64.1 | 63.8 | 59.0 | 69.1 | **71.4** | 71.0 | 64.0 | 64.4 | 64.8 | 64.0 | 58.3 | 69.0 | **72.4** |
|  | *Group Avg.* | 78.1 | 71.3 | 71.4 | 71.1 | 71.3 | 66.9 | 73.8 | **77.0** | 76.2 | 69.7 | 69.8 | 69.8 | 69.7 | 65.9 | 72.4 | **75.3** |

## 4.4. Prospective Evaluation

To validate real-world utility, we evaluated our model on prospective cohorts from both CU and WCM (Table 2). HyperCT demonstrates strong generalization, achieving the highest average AUCs of 77.8% (CU) and 78.6% (WCM). Interestingly, while Single-Task Learning (STL) models perform comparably well on the retrospective study, they are consistently surpassed by HyperCT in this prospective setting. This suggests that multi-task frameworks learn more robust representations that better withstand the distributional shifts common in real-world deployment, a quality at which HyperCT's dynamic architecture excels.

Table 2: Comprehensive comparison of AUC scores (%) on prospective study. Best results among MTL methods are **bolded**, and second-best results are underlined.

| | Task | CU Prospective | | | | | | | | WCM Prospective | | | | | | | |
|---|---|---|---|---|---|---|---|---|---|---|---|---|---|---|---|---|---|
| | | STL | EW | UW | RLW | DWA | GLS | MGDA | HyperCT | STL | EW | UW | RLW | DWA | GLS | MGDA | HyperCT |
| | *Overall Average* | 75.4 | 69.7 | 70.9 | 71.6 | 70.9 | 66.8 | 74.4 | **77.8** | 75.2 | 73.0 | 72.9 | 73.2 | 73.0 | 68.1 | 76.1 | **78.6** |
| Opportunistic | Red. RV | 77.0 | 74.8 | 73.8 | 73.6 | 73.9 | 68.7 | 75.1 | **77.2** | 80.6 | 79.7 | 79.5 | 78.5 | 79.7 | 76.2 | **79.9** | 79.2 |
| | Red. LV | 74.1 | 70.5 | 71.1 | 70.8 | 71.1 | 65.4 | 74.1 | **76.6** | 72.5 | 70.0 | 69.7 | 69.5 | 70.0 | 66.0 | 73.2 | **77.2** |
| | Pulm. HTN | 71.7 | 70.1 | 70.9 | 71.8 | 71.1 | 69.9 | 72.7 | **73.6** | 71.0 | 72.9 | 72.5 | 73.2 | 73.0 | 68.7 | **75.7** | 75.5 |
| | Atrial Enlg. | 80.2 | 74.1 | 72.0 | 72.3 | 72.0 | 68.6 | 74.3 | **80.0** | 83.5 | 77.1 | 77.1 | 77.7 | 77.1 | 69.8 | 80.2 | **80.8** |
| | Vent. Enlg. | 78.3 | 61.0 | 71.5 | 73.3 | 71.2 | 65.7 | 77.0 | **86.6** | 67.0 | 69.5 | 69.4 | 71.0 | 69.5 | 63.8 | 73.7 | **81.8** |
| | LA Pressure | 76.8 | 73.7 | 75.7 | 75.8 | 75.9 | 73.4 | 76.8 | **78.5** | 77.7 | 76.2 | 76.0 | 75.9 | 76.2 | 72.3 | 77.9 | **79.1** |
| | RA Pressure | 70.0 | 64.0 | 61.5 | 63.7 | 61.0 | 56.2 | 70.6 | **72.3** | 74.2 | 65.4 | 65.8 | 66.4 | 65.4 | 59.7 | 72.4 | **76.7** |

## 4.5. Clinical Utility: Decision Curve Analysis

To demonstrate clinical utility beyond discriminative performance, we performed Decision Curve Analysis (DCA) (Vickers and Elkin, 2006) for all 7 opportunistic cardiac tasks. DCA quantifies *net benefit*—the trade-off between true positives and false positives—across decision thresholds, directly measuring clinical value. For opportunistic cardiac screening, without a predictive model clinicians face two suboptimal strategies: refer all CT patients for echocardiography ("treat all"—costly with low yield) or refer none ("treat none"—missed diagnoses). A model provides clinical value when its curve lies above both baselines.

Figure 2 shows DCA for all 7 opportunistic tasks on the CU prospective cohort. HyperCT consistently demonstrates positive net benefit over threshold ranges of 5-80% across all tasks. Notably, for high-impact conditions such as *Ventricular Enlargement* and *Atrial Enlargement*, HyperCT maintains substantial net benefit even at high thresholds (60-80%), indicating robust clinical utility for selective referral decisions. The curves for functional assessments (*Reduced LV/RV Systolic Function*) show consistent positive net benefit across the full threshold range, suggesting HyperCT can effectively triage patients who would benefit from echocardiographic evaluation of cardiac function.

To validate generalization of clinical utility across institutions, Figure 3 presents DCA for the WCM prospective cohort. Despite being an external institution with potentially different patient populations and imaging protocols, HyperCT maintains consistent positive net benefit across all tasks. This multi-center validation is critical for demonstrating that the clinical utility of HyperCT is not institution-specific but generalizes to real-world deployment scenarios. Full DCA results for retrospective cohorts are provided in Appendix Sec. K.

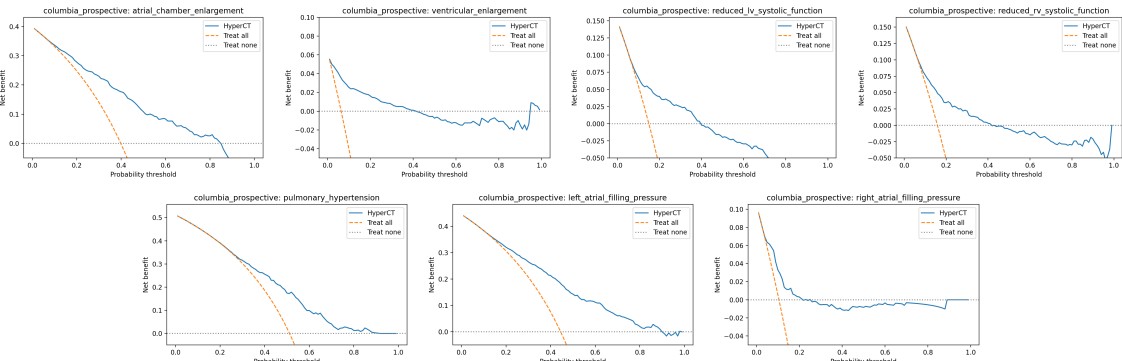

Figure 2: Decision Curve Analysis on CU prospective cohort for all 7 opportunistic cardiac tasks. HyperCT (blue) shows positive net benefit above "treat all" (orange) and "treat none" (gray) baselines across clinically relevant thresholds (5-80%).

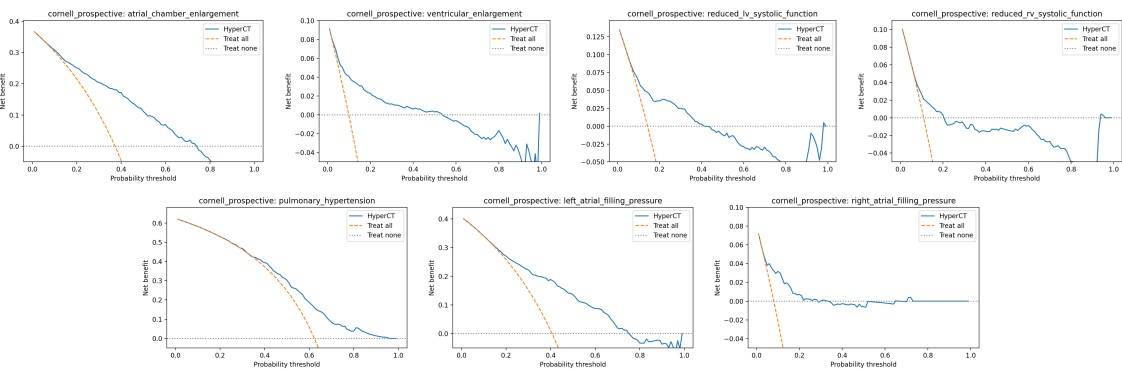

Figure 3: Decision Curve Analysis on WCM prospective cohort (external validation). HyperCT demonstrates consistent clinical utility across institutions, with positive net benefit maintained for all 7 opportunistic tasks.

## 5. Ablation Study

Table 3: Comparison of AUC scores (%) for LoRA module variants on retrospective study. Best results are **bolded**, and second best are underlined. Full table is in Appendix Sec. A

| | Method | Params | Med. Mat. | Art. Calc. | Cardiomeg. | Peri. Eff. | Cor. Art. Calc. | Hiatal Hernia | Lymphadenop. | Emphysema | Atelectasis | Lung Nodule | Lung Opacity | Pulm. Fibrosis | Pleural Eff. | Mosaic Attn. | Peribronchial | Consolidation | Bronchiectasis | Septal Thick. | Avg |
|---|---|---|---|---|---|---|---|---|---|---|---|---|---|---|---|---|---|---|---|---|---|
| CU | Attn Only | 35.9M | 85.3 | 81.8 | 86.7 | **68.8** | **88.3** | 67.1 | **67.4** | 77.7 | 77.0 | **70.4** | 77.4 | 84.2 | 95.3 | 70.3 | 65.6 | 85.1 | 80.1 | 75.4 | 77.9 |
| | MLP only | 49.3M | 85.0 | 81.6 | **87.1** | 67.0 | 87.9 | 66.7 | 66.9 | 78.0 | 76.9 | 69.8 | 77.7 | 85.0 | 95.2 | 70.0 | 65.5 | 85.4 | **80.6** | 75.1 | 77.8 |
| | HyperCT | 85.0M | **85.8** | **81.9** | 87.0 | 68.5 | 88.2 | 67.6 | 67.1 | **79.1** | **77.7** | 70.0 | **78.2** | **85.2** | **95.6** | **71.6** | **66.3** | **86.3** | **80.6** | **75.8** | **78.5** |
| WCM | Attn Only | 35.9M | 87.1 | **76.0** | 86.4 | **71.2** | 82.1 | 67.0 | 69.4 | 73.5 | **78.4** | **64.8** | 76.9 | 80.7 | 95.6 | 65.7 | 65.3 | 80.7 | 76.5 | 79.0 | 76.5 |
| | MLP only | 49.3M | 86.8 | 75.7 | **87.2** | 70.2 | 82.0 | 65.4 | **69.5** | 73.9 | 77.2 | 64.4 | 77.2 | **81.3** | 95.6 | 66.3 | 65.1 | 80.4 | 76.6 | 78.8 | 76.3 |
| | HyperCT | 85.0M | **87.3** | **76.0** | 87.0 | 71.1 | **82.3** | **68.8** | 69.4 | 74.9 | 78.2 | 64.4 | **78.0** | 80.6 | **95.9** | **68.1** | **66.5** | **82.0** | **76.8** | **79.3** | **77.0** |

**Module selection.** Table 3 presents an ablation study comparing the AUC scores of three LoRA module variants—Attn Only, MLP only, and HyperCT—across 18 conventional med-

ical imaging tasks on the CU and WCM retrospective study. The results demonstrate that the HyperCT architecture (85.0M parameters) consistently delivers the superior performance, achieving the highest average AUC scores of 78.5% for the CU group and 77.0% for the WCM group. While the lighter Attn Only (35.9M) and MLP only (49.3M) variants perform comparably to one another with slightly lower averages, HyperCT secures the top results (bolded) in the vast majority of individual pathologies, such as Emphysema, Consolidation, and Septal Thickening, across both datasets.

Table 4: Ablation of LoRA Rank ($r$) dimensions on retrospective study (Conversional labels). Best results are **bolded**, and second-best results are underlined. Full table is in Appendix. Sec. D

| | Rank | Med. Mat. | Art. Calc. | Cardiomeg. | Peri. Eff. | Cor. Art. Calc. | Hiatal Hernia | Lymphadenop. | Emphysema | Atelectasis | Lung Nodule | Lung Opacity | Pulm. Fibrosis | Pleural Eff. | Mosaic Attn. | Peribronchial | Consolidation | Bronchiectasis | Septal Thick. | Avg |
|---|---|---|---|---|---|---|---|---|---|---|---|---|---|---|---|---|---|---|---|---|
| **CU** | $r=1$ | 76.0 | 80.2 | 85.4 | 66.1 | 84.8 | 65.8 | 65.8 | 73.9 | 74.5 | 69.8 | 75.7 | 82.3 | 94.5 | 66.6 | 64.0 | 81.1 | 78.9 | 74.2 | 75.2 |
| | $r=2$ | 83.6 | 80.7 | 85.6 | 66.8 | 86.2 | 66.6 | **67.5** | 75.5 | 76.6 | 69.8 | 77.1 | 84.3 | 94.9 | 69.6 | 65.0 | 83.3 | 79.9 | 75.3 | 76.7 |
| | $r=4$ | 84.8 | 81.1 | 86.7 | **68.5** | 87.1 | 65.4 | 67.4 | 76.8 | 76.6 | 70.2 | 77.4 | 84.1 | 95.2 | 70.5 | 65.6 | 84.9 | 80.3 | 75.1 | 77.4 |
| | $r=8$ | 85.6 | 81.5 | 86.9 | 68.4 | 87.7 | 66.0 | 67.3 | 77.0 | 76.9 | 70.4 | 77.6 | 84.5 | 95.3 | 70.3 | 65.5 | 85.2 | 80.3 | 75.4 | 77.7 |
| | $r=16$ | **85.8** | **81.9** | **87.0** | **68.5** | **88.2** | 67.6 | 67.1 | **79.1** | **77.7** | 70.0 | **78.2** | **85.2** | **95.6** | **71.6** | 66.3 | **86.3** | 80.6 | **75.8** | **78.5** |
| **WCM** | $r=1$ | 76.6 | 74.5 | 84.8 | 69.1 | 79.1 | 62.8 | 68.2 | 70.1 | 75.7 | 63.9 | 74.6 | 80.3 | 94.6 | 62.2 | 63.5 | 76.3 | 75.0 | 77.8 | 73.5 |
| | $r=2$ | 85.8 | 75.4 | 85.4 | 69.0 | 80.8 | 63.8 | 69.4 | 71.4 | 77.0 | 64.0 | 75.9 | **81.5** | 95.1 | 64.3 | 63.7 | 78.5 | 76.3 | 78.3 | 74.8 |
| | $r=4$ | 86.7 | 75.6 | 86.6 | **71.2** | 81.4 | 63.9 | 69.5 | 73.9 | 77.2 | 64.3 | 76.7 | 80.7 | 95.5 | 66.7 | 64.9 | 80.2 | 76.2 | 78.5 | 75.8 |
| | $r=8$ | 86.9 | 75.8 | 86.4 | 70.9 | 82.0 | 64.6 | 69.3 | 72.6 | 77.4 | 64.5 | 77.1 | 81.1 | 95.7 | 65.5 | 65.0 | 80.5 | 75.5 | 78.8 | 75.9 |
| | $r=16$ | **87.3** | **76.0** | **87.0** | 71.1 | **82.3** | 68.8 | 69.4 | 74.9 | 78.2 | 64.4 | 78.0 | 80.6 | 95.9 | 68.1 | 66.5 | 82.0 | 76.8 | 79.3 | **77.0** |

**LoRA rank.** Table 4 presents the ablation study on the impact of the LoRA rank dimension ($r$) across 18 conventional tasks on the retrospective study. We observe a consistent trend where increasing the rank from $r = 1$ to $r = 16$ yields performance gains across both institutions. Specifically, the configuration with $r = 16$ achieves the highest average AUC scores of 78.5% for Columbia (CU) and 77.0% for Cornell (WCM), securing the best results in the majority of individual tasks. This indicates that while Low-Rank Adaptation is designed for parameter efficiency, a sufficient rank dimension is essential to provide the necessary model capacity for effectively adapting the frozen backbone features to a diverse range of cardiopulmonary pathologies.

**Backbone selection.** Table 5 evaluates the impact of backbone selection by benchmarking the 3D-pretrained CTViT (Hamamci et al., 2023) against the 2D-pretrained DINOv3 foundation model on conventional radiological tasks. The results unequivocally favor the 2D backbone, with DINOv3 establishing a new baseline by outperforming CTViT across every individual task in both the CU and WCM test sets. This shows the importance of backbone selection for base model. With DINOv3 extensively pretrained on large-scale 2D natural images, it appears to capture more generalizable features that transfer effectively to medical imaging tasks, even when applied to 3D volumetric data through slice-wise processing. This finding underscores the potential of leveraging large-scale 2D pretraining for enhancing performance in 3D medical imaging applications.

**Task visualization.** Fig. 4 visualizes the Principal Component Analysis (PCA) of the task-specific LoRA weights generated by the hypernetwork. A distinct semantic separa-

Table 5: Comparison of 3D (CTViT) and 2D (DINOv3) backbones on retrospective study (Conventional labels). Best results are **bolded**. Full table is in Appendix. C

| | Method | Med. Mat. | Art. Calc. | Cardiomeg. | Peri. Eff. | Cor. Art. Calc. | Hiatal Hernia | Lymphadenop. | Emphysema | Atelectasis | Lung Nodule | Lung Opacity | Pulm. Fibrosis | Pleural Eff. | Mosaic Attn. | Peribronchial | Consolidation | Bronchiectasis | Septal Thick. | Avg |
|---|---|---|---|---|---|---|---|---|---|---|---|---|---|---|---|---|---|---|---|---|
| CU | CTViT | 66.7 | 68.3 | 79.4 | 67.0 | 71.7 | 65.8 | 65.5 | 71.3 | 71.5 | 69.0 | 71.8 | 76.8 | 91.0 | 67.5 | 61.6 | 77.9 | 74.4 | 68.6 | 71.1 |
| CU | DINOv3 | **85.8** | **81.9** | **87.0** | **68.5** | **88.2** | **67.6** | **67.1** | **79.1** | **77.7** | **70.0** | **78.2** | **85.2** | **95.6** | **71.6** | **66.3** | **86.3** | **80.6** | **75.8** | **78.5** |
| WCM | CTViT | 65.8 | 64.3 | 79.1 | 69.1 | 67.9 | 65.3 | 67.5 | 64.4 | 72.3 | 62.8 | 69.9 | 69.9 | 89.9 | 63.4 | 59.4 | 72.0 | 67.1 | 70.3 | 68.3 |
| WCM | DINOv3 | **87.3** | **76.0** | **87.0** | **71.1** | **82.3** | **68.8** | **69.4** | **74.9** | **78.2** | **64.4** | **78.0** | **80.6** | **95.9** | **68.1** | **66.5** | **82.0** | **76.8** | **79.3** | **77.0** |

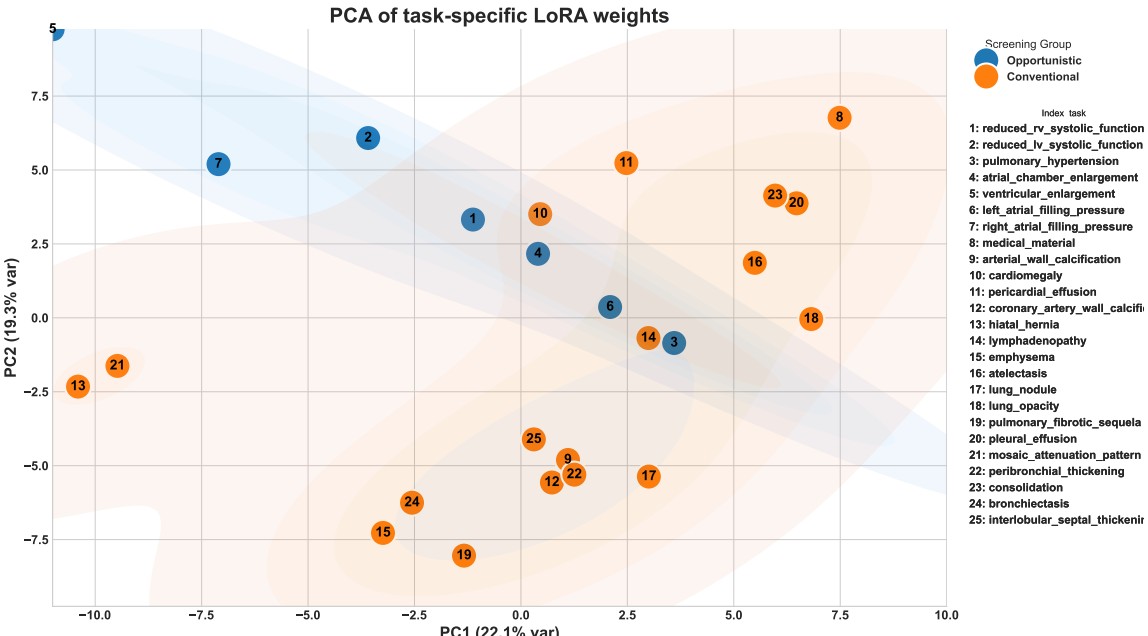

Figure 4: Principle Component Analysis (PCA) of task-specific LoRA. Blue is opportunistic labels and orange is conventional labels. Number is the index of labels.

tion is evident between the **Opportunistic** (blue) and **Conventional** (orange) screening groups; the opportunistic tasks—primarily relating to cardiac function and hemodynamics—cluster in a specific region separate from the broader distribution of conventional radiological findings. Note that index 10 (Cardiomegaly) and 14 (lymphadenopathy) are overlap with the blue manifold because they are associated with cardiovascular health. This clustering suggests that the hypernetwork effectively captures the underlying domain shifts between these task categories, automatically learning to allocate different parameter subspaces to address the distinct feature extraction requirements of physiological estimation versus anatomical detection. We also provide a quantitative clustering analysis is provided in Appendix Sec. J.

**Saliency map.** Fig. 5 illustrates the model's visual attention through Grad-CAM-generated saliency maps for a range of diagnostic tasks, which are divided into opportunistic cardio-

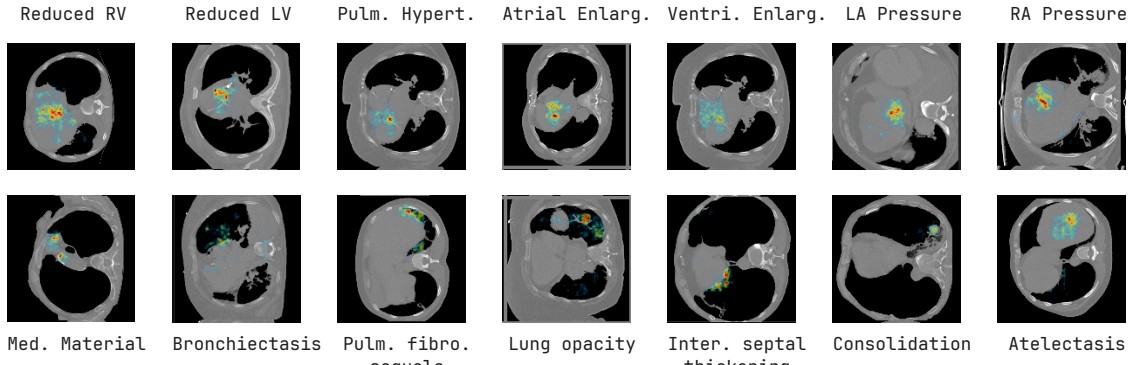

Figure 5: Saliency maps generated using Grad-CAM for different tasks. First row is opportunistic screening tasks and second row is part of the conventional screening tasks.

vascular screenings (top row) and conventional pulmonary screenings (bottom row). For opportunistic tasks, the saliency maps consistently and accurately localize the attention regions within the cardiac silhouette. Similarly, for conventional pulmonary findings, the model correctly focuses its attention on the relevant areas within the lung parenchyma and pleura. This strong alignment between the model's focus and the expected anatomical locations for each specific pathology enhancing the interpretability and trustworthiness of its predictions.

## 6. Conclusion

In this work, we introduced HyperCT, a novel framework using a LoRA-integrated hypernetwork to unify conventional and opportunistic chest CT screening. Our model demonstrated superior generalization on prospective, multi-institutional data, outperforming both strong MTL baselines and matching with specialized Single-Task Learning models. Analyses of the generated LoRA weights and saliency maps confirmed that our dynamic approach learns a meaningful, task-adaptive parameter space. HyperCT offers a parameter-efficient and unified solution for holistic patient assessment, paving the way for maximizing the clinical value of routine medical imaging.

**Limitations and Future Work.** Scalability to additional tasks depends on their relationship to existing tasks. For related tasks (e.g., additional cardiac or pulmonary findings), adding a new task requires only learning a new task embedding while the hypernetwork parameters remain fixed. However, for anatomically unrelated tasks (e.g., osteoporosis, sarcopenia), joint retraining may be required as the current model is optimized for cardiopulmonary features. Additionally, we currently use equal task weighting; exploring advanced task weighting or sampling strategies may further improve performance on tasks with limited labels. Developing a more general hypernetwork that transfers across anatomical domains is an interesting direction for future work.

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

# Appendix A. Full Table for Ablation LoRA Module

Table 6: Ablation Study: Comparison of AUC scores (%) for LoRA Module variants on Retrospective study. Best results are bolded.

|  | Task | CU Test | | | WCM Test | | |
|---|---|---|---|---|---|---|---|
|  |  | Attn Only | MLP only | HyperCT | Attn Only | MLP only | HyperCT |
|  | *Overall Average* | 77.8 | 77.7 | **78.1** | 76.1 | 76.1 | **76.5** |
| Conventional | Medical Material | 85.3 | 85.0 | **85.8** | 87.1 | 86.8 | **87.3** |
| | Arterial Wall Calcification | 81.8 | 81.6 | **81.9** | **76.0** | 75.7 | **76.0** |
| | Cardiomegaly | 86.7 | **87.1** | 87.0 | 86.4 | **87.2** | 87.0 |
| | Pericardial Effusion | **68.8** | 67.0 | 68.5 | **71.2** | 70.2 | 71.1 |
| | Coronary Artery Wall Calc. | **88.3** | 87.9 | 88.2 | 82.1 | 82.0 | **82.3** |
| | Hiatal Hernia | 67.1 | 66.7 | **67.6** | 67.0 | 65.4 | **68.8** |
| | Lymphadenopathy | **67.4** | 66.9 | 67.1 | 69.4 | **69.5** | 69.4 |
| | Emphysema | 77.7 | 78.0 | **79.1** | 73.5 | 73.9 | **74.9** |
| | Atelectasis | 77.0 | 76.9 | **77.7** | **78.4** | 77.2 | 78.2 |
| | Lung Nodule | **70.4** | 69.8 | 70.0 | 64.8 | 64.4 | 64.4 |
| | Lung Opacity | 77.4 | 77.7 | **78.2** | 76.9 | 77.2 | **78.0** |
| | Pulmonary Fibrotic Sequela | 84.2 | 85.0 | **85.2** | 80.7 | **81.3** | 80.6 |
| | Pleural Effusion | 95.3 | 95.2 | **95.6** | 95.6 | 95.6 | **95.9** |
| | Mosaic Attenuation Pattern | 70.3 | 70.0 | **71.6** | 65.7 | 66.3 | **68.1** |
| | Peribronchial Thickening | 65.6 | 65.5 | **66.3** | 65.3 | 65.1 | **66.5** |
| | Consolidation | 85.1 | 85.4 | **86.3** | 80.7 | 80.4 | **82.0** |
| | Bronchiectasis | 80.1 | **80.6** | **80.6** | 76.5 | 76.6 | **76.8** |
| | Interlobular Septal Thick. | 75.4 | 75.1 | **75.8** | 79.0 | 78.8 | **79.3** |
| | *Group Avg.* | 77.9 | 77.8 | **78.5** | 76.5 | 76.3 | **77.0** |
| Opportunistic | Reduced RV Systolic Function | 77.1 | 77.1 | **77.5** | 77.5 | **78.0** | 77.9 |
| | Reduced LV Systolic Function | **77.2** | 77.1 | 77.0 | **74.8** | 74.7 | 74.6 |
| | Pulmonary Hypertension | **72.9** | 72.6 | 72.7 | 71.9 | **72.2** | 72.0 |
| | Atrial Chamber Enlargement | 82.6 | 82.0 | **83.0** | 79.6 | **80.1** | 79.9 |
| | Ventricular Enlargement | 80.5 | **81.2** | 80.4 | 73.2 | **73.6** | 73.1 |
| | Left Atrial Filling Pressure | 77.0 | **77.1** | **77.1** | 77.0 | **77.1** | **77.1** |
| | Right Atrial Filling Pressure | 73.4 | **73.6** | 71.4 | 73.0 | **73.2** | 72.4 |
| | *Group Avg.* | **77.2** | **77.2** | 77.0 | 75.3 | **75.6** | 75.3 |

Table 7: Ablation Study: Comparison of AUC scores (%) for LoRA Module variants on Prospective Datasets. Best results are bolded.

|  | Task | CU Prospective | | | WCM Prospective | | |
|---|---|---|---|---|---|---|---|
|  |  | Attn Only | MLP only | HyperCT | Attn Only | MLP only | HyperCT |
|  | *Overall Average* | 77.7 | 77.5 | **77.8** | 78.9 | **79.6** | 78.6 |
| Opportunistic | Reduced RV Systolic Function | 76.6 | 76.6 | **77.2** | 78.7 | **80.5** | 79.2 |
| | Reduced LV Systolic Function | **76.9** | 76.8 | 76.6 | 79.0 | **80.7** | 77.2 |
| | Pulmonary Hypertension | 71.7 | 73.4 | **73.6** | 75.0 | **76.2** | 75.5 |
| | Atrial Chamber Enlargement | **80.1** | 79.5 | 80.0 | 81.6 | **82.1** | 80.8 |
| | Ventricular Enlargement | 85.8 | 86.2 | **86.6** | 81.7 | **83.3** | 81.8 |
| | Left Atrial Filling Pressure | 78.3 | 77.8 | **78.5** | 79.4 | **79.8** | 79.1 |
| | Right Atrial Filling Pressure | **74.5** | 72.5 | 72.3 | **76.8** | 74.7 | 76.7 |

## Appendix B. Valid label fraction

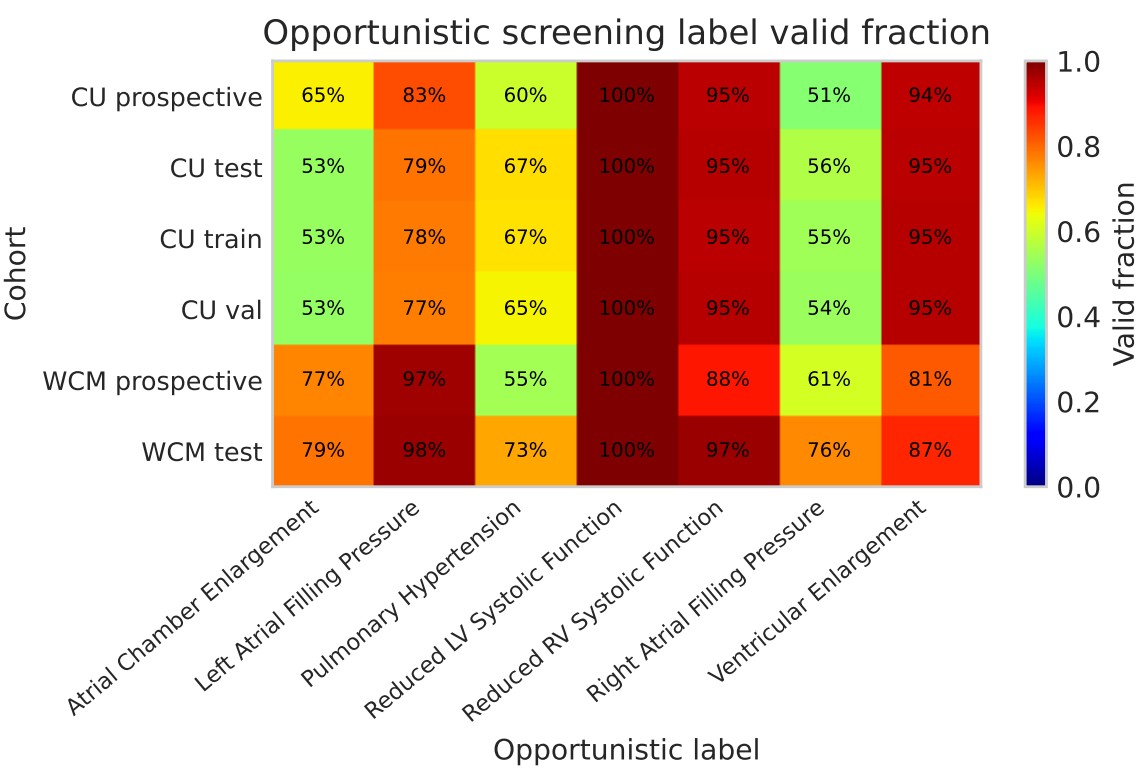

Figure 6: Sample valid fraction heatmaps for opportunistic screening labels

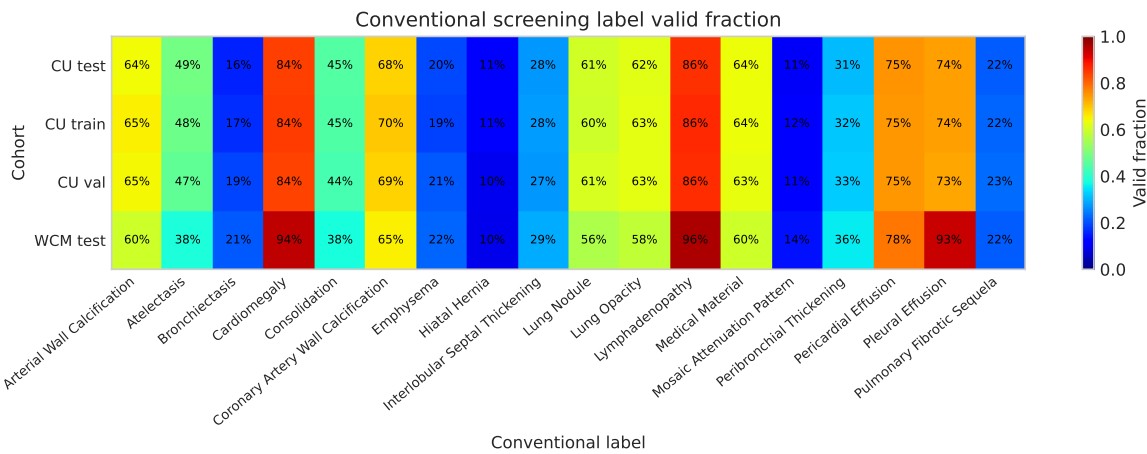

Figure 7: Sample valid fraction heatmaps for conventional screening labels

## Appendix C. Ablation: Backbone selection

Table 8: Comparison of 3D (CTViT) and 2D (DINOv3) backbones on Opportunistic Tasks across all datasets. Best results are bolded.

|  | Method | Reduced RV | Reduced LV | Pulmonary Hypertension | Atrial Enlargement | Ventricular Enlargement | LA Pressure | RA Pressure | Avg |
|---|---|---|---|---|---|---|---|---|---|
| CU (Retro) | CTViT-Encoder | 72.4 | 70.6 | 69.2 | 75.9 | 73.4 | 71.5 | 70.0 | 71.9 |
|  | DINOv3 | **77.5** | **77.0** | **72.7** | **83.0** | **80.4** | **77.1** | **71.4** | **77.0** |
| WCM (Retro) | CTViT-Encoder | 72.5 | 66.3 | 70.0 | 72.1 | 66.9 | 71.4 | 70.0 | 69.9 |
|  | DINOv3 | **77.9** | **74.6** | **72.0** | **79.9** | **73.1** | **77.1** | **72.4** | **75.3** |
| CU (Prosp) | CTViT-Encoder | 70.8 | 68.6 | 69.7 | 75.6 | 79.2 | 74.8 | 68.1 | 72.4 |
|  | DINOv3 | **77.2** | **76.6** | **73.6** | **80.0** | **86.6** | **78.5** | **72.3** | **77.8** |
| WCM (Prosp) | CTViT-Encoder | 74.0 | 70.7 | 73.8 | 73.0 | 74.5 | 73.7 | 74.3 | 73.4 |
|  | DINOv3 | **79.2** | **77.2** | **75.5** | **80.8** | **81.8** | **79.1** | **76.7** | **78.6** |

# Appendix D. Ablation: rank selection

Table 9: Ablation of LoRA Rank ($r$) dimensions on Opportunistic Tasks (Retrospective vs. Prospective Test Sets). Best results are bolded per institution.

| | Rank | Retrospective | | | | | | | | Prospective | | | | | | | |
| | | Red. RV Sys. | Red. LV Sys. | Pulm. HTN | Atrial Enl. | Vent. Enl. | LA Pressure | RA Pressure | **Avg** | Red. RV Sys. | Red. LV Sys. | Pulm. HTN | Atrial Enl. | Vent. Enl. | LA Pressure | RA Pressure | **Avg** |
|---|---|---|---|---|---|---|---|---|---|---|---|---|---|---|---|---|---|
| **CU** | $r=1$ | 75.4 | 75.4 | 72.0 | 81.3 | 67.9 | 75.6 | 71.8 | 74.2 | 75.2 | 73.8 | 70.9 | 77.4 | 71.2 | 76.1 | 68.6 | 73.3 |
| | $r=2$ | 76.2 | 75.6 | 71.9 | 81.1 | 77.2 | 75.2 | 72.4 | 75.6 | 75.4 | 74.7 | 71.9 | 78.0 | 79.0 | 76.1 | **73.6** | 75.5 |
| | $r=4$ | **76.7** | 76.8 | **72.4** | 82.2 | **80.7** | 76.6 | 72.9 | 76.9 | 76.1 | **76.6** | 71.8 | **79.2** | 85.6 | 77.3 | 73.3 | 77.1 |
| | $r=8$ | 76.6 | **77.1** | **72.4** | **82.6** | 80.3 | **76.9** | **73.7** | **77.1** | **76.2** | 76.3 | **72.1** | 79.0 | **85.8** | **77.5** | 73.6 | **77.2** |
| **WCM** | $r=1$ | 76.0 | 72.4 | 70.9 | 78.5 | 64.6 | 76.0 | 71.0 | 72.8 | 78.4 | 77.6 | 73.2 | 79.9 | 75.0 | 77.8 | 73.2 | 76.4 |
| | $r=2$ | 77.4 | 72.9 | 71.0 | 78.5 | 68.8 | 75.4 | 71.5 | 73.6 | **79.0** | **79.8** | 75.0 | 80.0 | 79.9 | 78.1 | 73.5 | 77.9 |
| | $r=4$ | **78.0** | 74.5 | **72.1** | **79.5** | 72.2 | 76.5 | 73.2 | 75.1 | 78.2 | 79.7 | 75.4 | **81.6** | 81.9 | 79.2 | **76.2** | 78.9 |
| | $r=8$ | 77.4 | **74.6** | 71.9 | **79.5** | **73.0** | **76.6** | **73.4** | **75.2** | 78.8 | 79.5 | **75.5** | 80.9 | **82.5** | **79.5** | 76.0 | **79.0** |

## Appendix E. Theoretical Analysis of Parameter Efficiency

To analysis the parameter efficiency of introducing LoRA, we analyze the complexity of the hypernetwork with respect to the width of a ViT. Let the ViT base model consists of layers with a hidden embedding dimension of $D$. A standard weight matrix $\mathbf{W}_m$ (e.g. either multi-head attention or Feed Forward Netowrk (FFN)) typically has dimensions $\mathbf{W}_m \in \mathbb{R}^{D \times D}$. Let $d_h$ be the dimension of the hypernetwork hidden layer.

*1) Naive Full-Rank Generation (Quadratic Complexity):* In a direct regression scheme, the final projection layer of $h_\phi$ must output a flattened weight matrix of size $D^2$. The number of parameter denoted as $P_{\text{full}}$ is given by:

$$P_{\text{full}} = d_h \cdot D^2 = \mathcal{O}(D^2) \tag{2}$$

This shows that full-rank requires quadratic complexity with respect to the ViT hidden dimension $D$.

*2) Low-Rank Adaptation Generation (Linear Complexity):* By adopting LoRA, $h_\phi$ by-passes the generatioin of full matrix $\mathbf{W}_m$. Instead, it generates two low-rank matrices $\mathbf{A}_m$ and $\mathbf{B}_m$ with dimensions $\mathbf{A}_m \in \mathbb{R}^{r \times D}$ and $\mathbf{B}_m \in \mathbb{R}^{D \times r}$. The number of parameters $P_{\text{LoRA}}$ in this case is:

$$P_{\text{LoRA}} = d_h \times D \times 2r = \mathcal{O}(D) \tag{3}$$

since $r$ is fixed and $r \ll D$. The complexity is now linear with respect to $D$. This analysis demonstrates that integrating LoRA into the hypernetwork architecture reduces the parameter complexity from quadratic to linear with respect to the ViT hidden dimension $D$. This significant reduction enables the practical deployment of hypernetwork-based multi-task learning in large-scale medical imaging applications.

## Appendix F. Ablation: Task Sampling Strategy

Table 10: Ablation Study: Comparison of task sampling strategies. Average AUC (%) across all 25 tasks. Results show minimal difference between strategies, indicating robustness to sampling choice.

| Sampling Strategy | CU Retro | WCM Retro | CU Prosp | WCM Prosp |
|---|---|---|---|---|
| Uniform Random (ours) | **78.1** | **76.5** | **77.8** | **78.6** |
| Inverse-Prevalence Weighted | 77.9 | 76.5 | 77.5 | 78.3 |

## Appendix G. Bootstrap Confidence Intervals

To quantify uncertainty, we computed bootstrap 95% confidence intervals (1000 iterations) for HyperCT. Tables 11 and 12 report CIs for retrospective and prospective cohorts respectively. CI widths are notably tighter for retrospective evaluation due to larger sample sizes.

**Retrospective Evaluation.** Table 11 shows CIs for all 25 tasks on retrospective test sets.

Table 11: Bootstrap 95% CIs for HyperCT on retrospective evaluation. AUC (%) with 95% CI over 1000 iterations.

| Task | AUC (95% CI) | | Sample Size | |
| --- | --- | --- | --- | --- |
| | **CU Retro** | **WCM Retro** | **CU** | **WCM** |
| ***Opportunistic (Cardiology)*** | | | | |
| Reduced RV Systolic Function | 76.7 (75.0–78.3) | 77.6 (76.2–79.1) | 4,930 | 7,899 |
| Reduced LV Systolic Function | 77.0 (75.3–78.7) | 74.6 (73.1–76.1) | 5,174 | 8,110 |
| Pulmonary Hypertension | 72.0 (70.2–73.5) | 71.5 (70.0–73.0) | 3,498 | 5,935 |
| Atrial Chamber Enlargement | 82.7 (81.1–84.1) | 80.0 (78.9–81.0) | 2,748 | 6,372 |
| Ventricular Enlargement | 80.9 (78.4–83.3) | 72.0 (70.4–73.7) | 4,911 | 7,032 |
| Left Atrial Filling Pressure | 76.7 (75.2–78.1) | 77.0 (76.0–78.0) | 4,058 | 7,918 |
| Right Atrial Filling Pressure | 71.5 (68.7–74.2) | 72.2 (70.4–74.0) | 2,921 | 6,193 |
| ***Conventional (Radiology)*** | | | | |
| Medical Material | 85.7 (84.7–86.7) | 87.5 (86.7–88.3) | 5,174 | 8,110 |
| Arterial Wall Calcification | 81.2 (79.9–82.4) | 75.9 (74.8–77.0) | 5,174 | 8,110 |
| Cardiomegaly | 86.8 (85.7–87.7) | 87.1 (86.2–88.0) | 5,174 | 8,110 |
| Pericardial Effusion | 67.3 (65.5–69.2) | 70.7 (69.2–72.3) | 5,174 | 8,110 |
| Coronary Artery Wall Calc. | 87.8 (86.8–88.8) | 82.6 (81.6–83.5) | 5,174 | 8,110 |
| Hiatal Hernia | 68.2 (65.7–70.6) | 68.6 (66.8–70.6) | 5,174 | 8,110 |
| Lymphadenopathy | 66.8 (65.4–68.4) | 69.0 (67.7–70.4) | 5,174 | 8,110 |
| Emphysema | 78.2 (76.6–79.9) | 74.1 (72.7–75.5) | 5,174 | 8,110 |
| Atelectasis | 77.0 (75.7–78.2) | 76.9 (75.9–78.0) | 5,174 | 8,110 |
| Lung Nodule | 69.8 (68.3–71.2) | 63.9 (62.7–65.1) | 5,174 | 8,110 |
| Lung Opacity | 77.8 (76.7–79.0) | 77.5 (76.4–78.4) | 5,174 | 8,110 |
| Pulmonary Fibrotic Sequela | 84.3 (82.8–85.8) | 79.4 (78.1–80.7) | 5,174 | 8,110 |
| Pleural Effusion | 95.0 (94.4–95.6) | 95.4 (95.0–95.9) | 5,174 | 8,110 |
| Mosaic Attenuation Pattern | 71.3 (69.0–73.7) | 68.1 (66.4–69.7) | 5,174 | 8,110 |
| Peribronchial Thickening | 65.4 (63.7–67.0) | 64.5 (63.2–65.7) | 5,174 | 8,110 |
| Consolidation | 85.9 (84.8–87.0) | 81.7 (80.6–82.7) | 5,174 | 8,110 |
| Bronchiectasis | 80.3 (78.5–81.8) | 76.6 (75.2–77.9) | 5,174 | 8,110 |
| Interlobular Septal Thickening | 75.2 (73.8–76.8) | 78.7 (77.5–79.9) | 5,174 | 8,110 |
| **Overall Average** | 77.7 (77.3–78.0) | 76.1 (75.9–76.4) | – | – |

**Prospective Evaluation.** Table 12 shows CIs for 7 opportunistic tasks on prospective test sets.

Table 12: Bootstrap 95% confidence intervals for HyperCT on prospective evaluation. AUC (%) with 95% CI computed over 1000 bootstrap iterations. CI width scales inversely with sample size.

| Task | AUC (95% CI) | | Sample Size | |
|---|---|---|---|---|
| | **CU** | **WCM** | **CU** | **WCM** |
| Reduced RV Systolic Function | 76.5 (73.2–79.6) | 78.5 (72.9–83.3) | 1,337 | 723 |
| Reduced LV Systolic Function | 76.7 (73.2–80.1) | 78.1 (73.6–82.3) | 1,411 | 817 |
| Pulmonary Hypertension | 73.5 (70.2–76.8) | 76.1 (71.6–80.8) | 842 | 449 |
| Atrial Chamber Enlargement | 80.2 (77.4–83.0) | 80.8 (77.4–84.3) | 921 | 628 |
| Ventricular Enlargement | 86.2 (82.4–90.1) | 81.4 (75.9–86.4) | 1,331 | 664 |
| Left Atrial Filling Pressure | 78.5 (75.9–81.0) | 78.8 (75.5–81.8) | 1,168 | 794 |
| Right Atrial Filling Pressure | 72.4 (67.4–77.2) | 77.7 (70.2–84.4) | 724 | 495 |
| **Overall Average** | 77.7 (76.4–79.1) | 78.8 (77.0–80.5) | – | – |

# Appendix H. LoRA Target Modules

Table 13 details the target modules for LoRA adaptation in HyperCT. We apply LoRA to all linear layers within the attention mechanism (Q, K, V projections and output projection) and the MLP block (fc1 up-projection and fc2 down-projection). For DINOv3 ViT-Base with 12 transformer blocks, this results in $M = 6 \times 12 = 72$ target modules. The hypernetwork generates separate LoRA weight matrices (A, B) for each module, conditioned on the task embedding and module positional embedding $\phi_{\text{pos}}$.

Table 13: LoRA target modules in HyperCT. All linear layers in attention and MLP blocks are adapted.

| Component | Layer | Per Block | Total (12 blocks) |
|---|---|---|---|
| Attention | Query (Q) projection | 1 | 12 |
| | Key (K) projection | 1 | 12 |
| | Value (V) projection | 1 | 12 |
| | Output projection | 1 | 12 |
| MLP | fc1 (up-projection) | 1 | 12 |
| | fc2 (down-projection) | 1 | 12 |
| **Total** | | **6** | **72** |

## Appendix I. Hypernetwork Architecture

Table 14 details the hypernetwork architecture in HyperCT. The hypernetwork processes concatenated task and module positional embeddings through a mixer and residual MLP blocks, then outputs LoRA weight matrices (A, B) via per-module heads. Default hyperparameters: latent size = 128, head input size = 512, LoRA rank = 16, dropout = 0.05.

Table 14: Hypernetwork architecture in HyperCT.

| Component | Architecture | Output Dim |
|---|---|---|
| Mixer | Linear → SiLU → Linear → SiLU | latent |
| MLP blocks (×2) | Residual(LayerNorm → Linear → SiLU → Linear → SiLU) | latent |
| Output projection | LayerNorm → Linear → SiLU → Linear → SiLU | head_in_size |
| Per-module heads | Linear | $r \times (d_{\text{in}} + d_{\text{out}})$ |

## Appendix J. Hierarchical Clustering of LoRA Weights

To quantitatively analyze learned task representations, we applied hierarchical clustering (Johnson, 1967) (complete linkage, cosine distance) to the flattened LoRA weight vectors, selecting the number of clusters $k$ by maximizing silhouette score (Rousseeuw, 1987) over $k \in \{2, \ldots, 8\}$. As shown in Fig. 8, this analysis yields **k=4 clusters with silhouette score 0.30**, revealing clinically interpretable groupings: (1) *Cardiac-Structural* tasks including calcifications and structural abnormalities; (2) *Cardiac-Functional* tasks relating to ventricular and atrial function; (3) *Acute Parenchymal* findings such as opacity, consolidation, and effusions; and (4) *Airway-Interstitial* diseases including bronchiectasis and fibrosis. Notably, tasks cluster by pathophysiology rather than label source—Cardiomegaly (extracted from radiology reports) groups with echocardiography-derived cardiac function tasks, not with other report-extracted findings. The emergence of anatomically coherent groupings from unsupervised clustering—without any clinical priors—validates that HyperCT learns anatomically meaningful specializations.

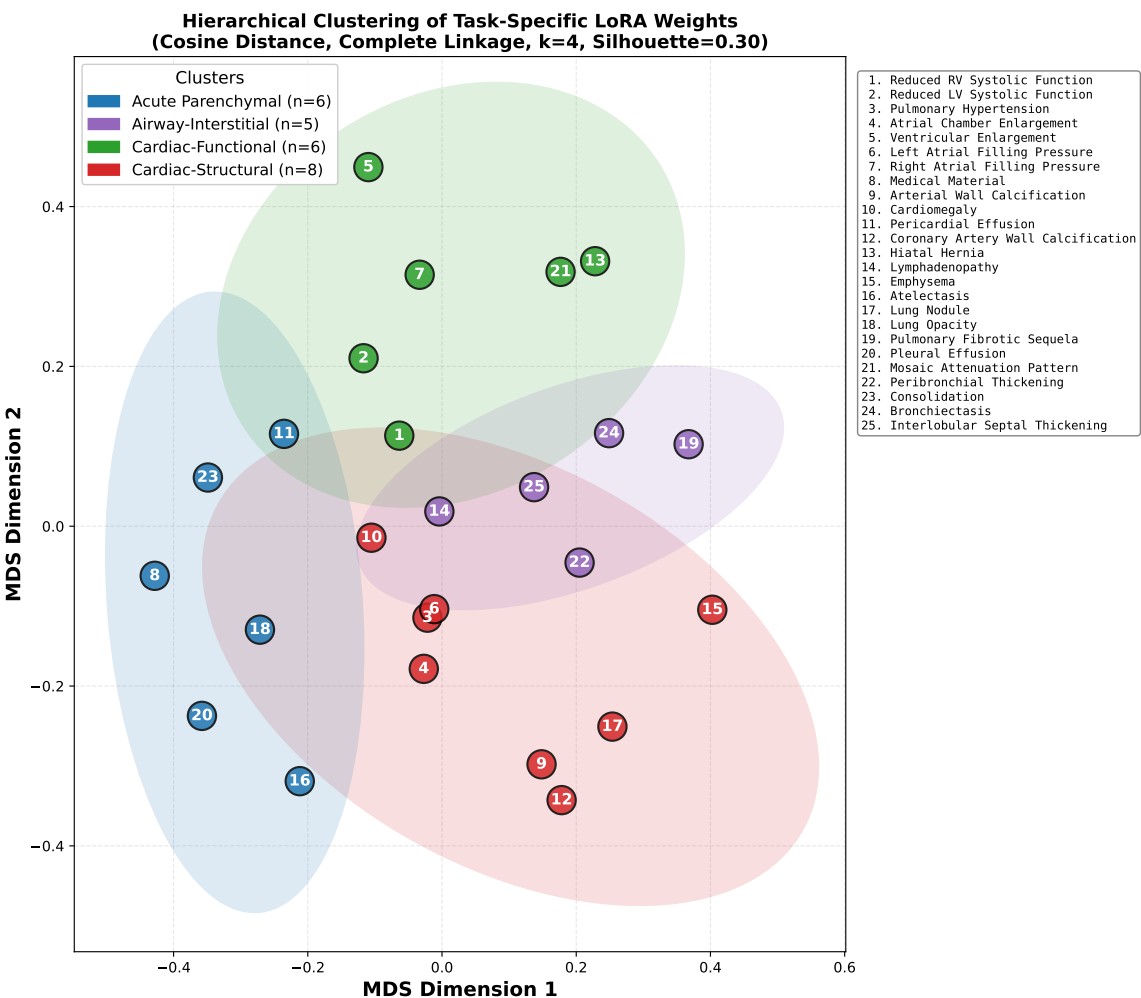

Figure 8: Hierarchical clustering of task-specific LoRA weights. MDS (Torgerson, 1952) projection of 25 tasks colored by cluster assignment (cosine distance, complete linkage, k=4, silhouette=0.30). Tasks naturally group into clinically interpretable categories without any clinical priors.

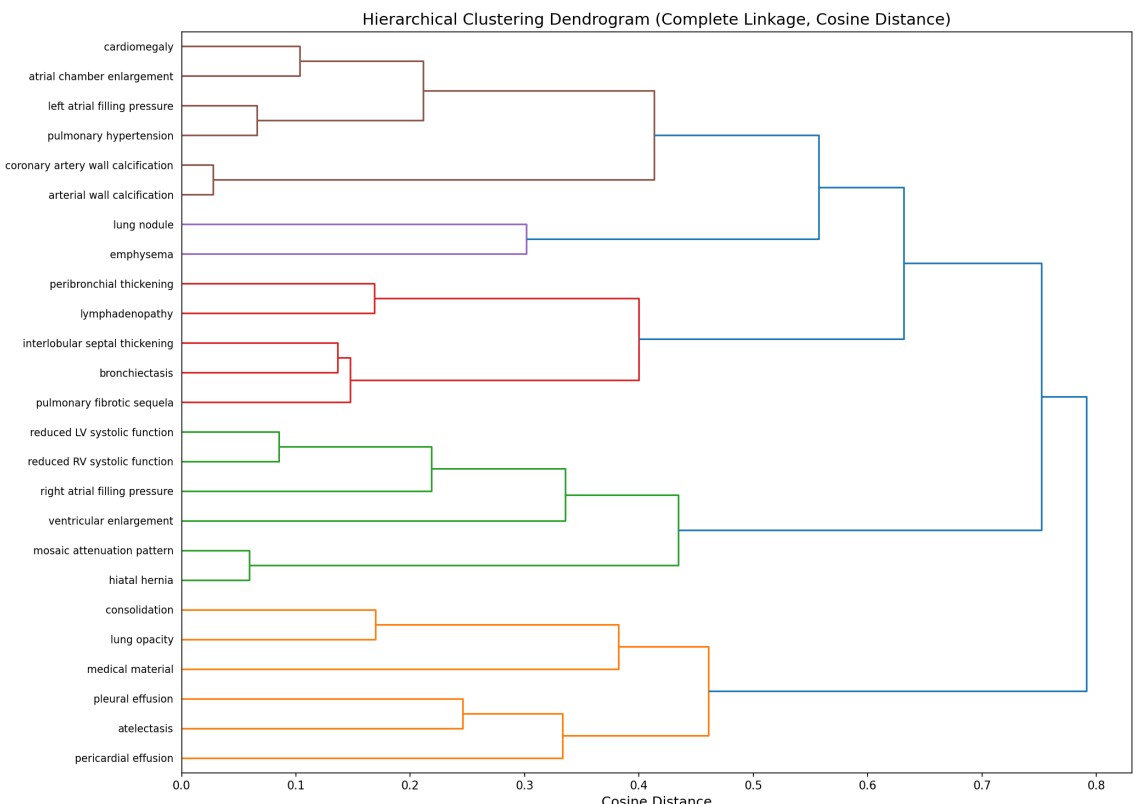

Figure 9: Dendrogram of hierarchical clustering (complete linkage, cosine distance) showing the tree structure of task relationships. Colors indicate the four identified clusters at the optimal cut point (k=4).

## Appendix K. Decision Curve Analysis (Retrospective Cohorts)

This section provides Decision Curve Analysis results for retrospective cohorts (prospective results shown in main text Sec. 4.2). Figures 10-11 show DCA curves for CU and WCM retrospective test sets across all 7 opportunistic cardiac tasks. Results consistently demonstrate positive net benefit, validating that clinical utility holds across both retrospective and prospective (main text) evaluation settings.

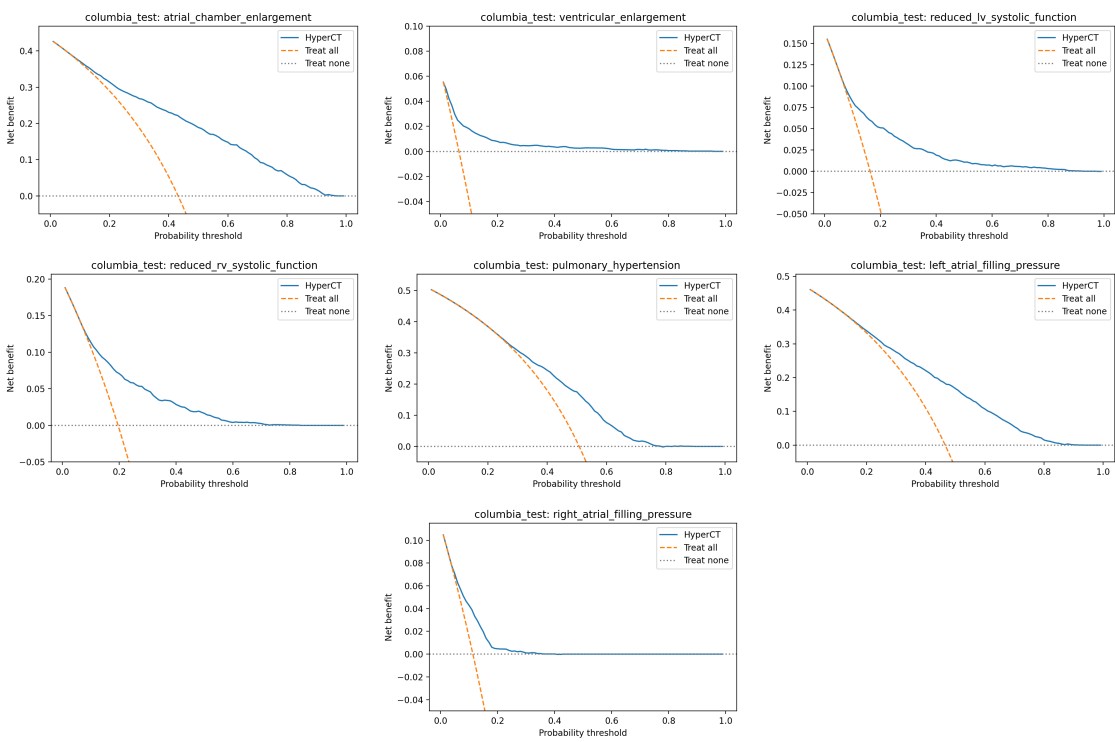

Figure 10: Decision Curve Analysis on CU retrospective cohort. HyperCT (blue) shows positive net benefit above "treat all" (orange) and "treat none" (gray) baselines.

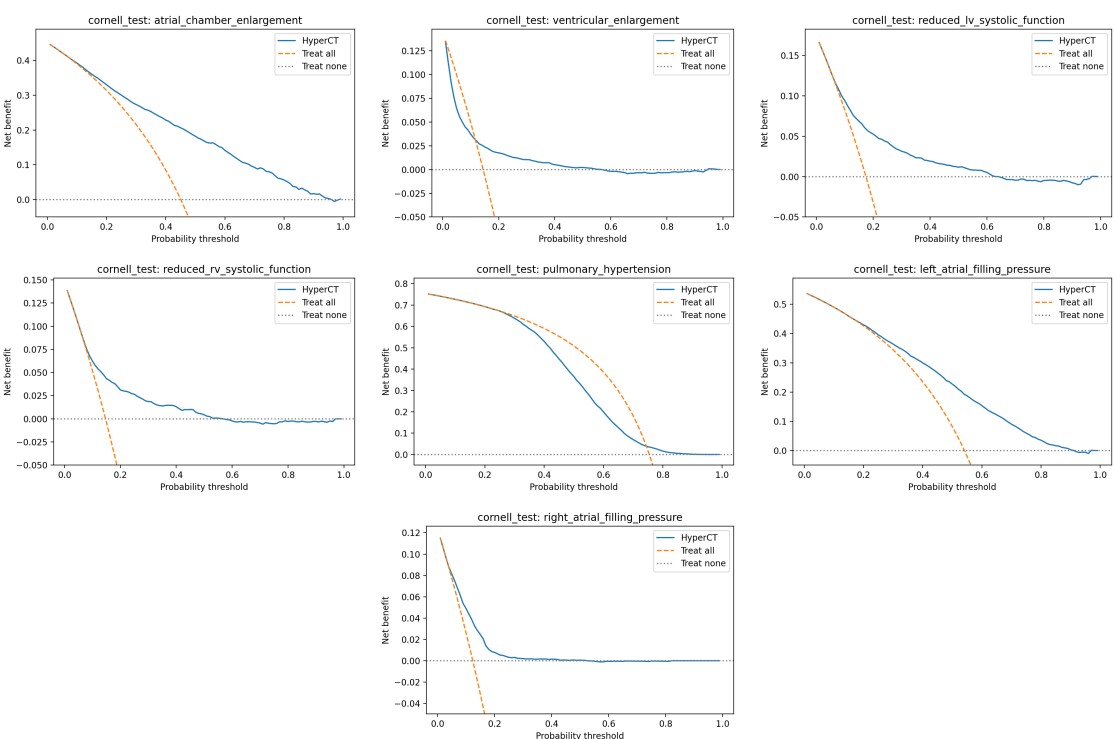

Figure 11: Decision Curve Analysis on WCM retrospective cohort.

## Appendix L. Opportunistic screening label curation

Table 15: Clinical Definitions for Cardiovascular Condition Curation

| Condition | Clinical Criteria for Presence |
| --- | --- |
| Left Atrial Filling Pressure (Elevated) | Considered present if **any** of the following echocardiographic findings are true:
• Mitral inflow E/A ratio $\geq 2$
• OR Indexed Left Atrial Volume (LAVI) $> 34$ mL/m$^2$
• OR Peak Tricuspid Regurgitation (TR) velocity $> 2.8$ m/s |
| Right Atrial Filling Pressure (Elevated) | Peak Tricuspid Regurgitation (TR) velocity $> 3.4$ m/s |
| Reduced Right Ventricular (RV) Systolic Function | Qualitative assessment of RV systolic function is anything other than 'normal' (e.g., 'mildly', 'moderately', or 'severely' reduced). |
| Reduced Left Ventricular (LV) Systolic Function | Left Ventricular Ejection Fraction (LVEF) $< 50\%$ |
| Pulmonary Hypertension | Estimated Pulmonary Artery Systolic Pressure (PASP) $> 35$ mmHg |
| Atrial Chamber Enlargement | Indexed Left Atrial Volume (LAVI) $> 34$ mL/m$^2$ |
| Ventricular Enlargement | Considered present if the Left Ventricular internal dimension in diastole (LVIDd) meets either of the following gender-specific criteria:
• For female patients: LVIDd $> 5.3$ cm
• OR for male patients: LVIDd $> 5.9$ cm |

## Appendix M. Conventional screening label prompt

**System Prompt: Radiology Expert Assistant**

""" You are a highly experienced radiology expert assistant.  Your task is to analyze a CT chest radiology report and determine, for each of the 18 abnormalities listed below, whether it is:
- Present (label 1):  The report explicitly describes or clearly implies the abnormality.  - Absent (label 0):  The report explicitly states that the abnormality is not present.  - Not mentioned (label -1):  There is no reference to the abnormality anywhere in the report.
Return your answer as a single JSON object that contains all 18 keys exactly as shown, with each value set to 1, 0, or -1.
Abnormalities and Their Definitions:
1.  "Medical material" Definition:  Any foreign medical objects or devices (e.g., central venous catheters, surgical clips, pacemakers, stents, fixation hardware).  Example Clues:  "central venous catheter present" or "surgical clips" indicate presence.
2.  "Arterial wall calcification" Definition:  Calcification along the walls of arteries, suggesting atherosclerotic changes.  Example Clues: Phrases like "atherosclerotic calcification" in arterial structures.
3.  "Cardiomegaly" Definition:  Enlargement of the heart silhouette. Example Clues:  "heart is enlarged" (present) or "borderline enlarged heart" (present) or "normal heart size" (absent).
4.  "Pericardial effusion" Definition:  Fluid accumulation within the pericardial sac.  Example Clues:  "pericardial effusion" or "small pericardial effusion" (present).
5.  "Coronary artery wall calcification" Definition:  Calcifications within the walls of the coronary arteries.  Example Clues: "calcification of the coronary vessels."
6.  "Hiatal hernia" Definition:  Protrusion of a portion of the stomach through the diaphragm into the chest cavity.  Example Clues:  Any mention of "hiatal hernia."
7.  "Lymphadenopathy" Definition:  Enlargement of lymph nodes (mediastinal, hilar, or axillary).  Example Clues:  "enlarged lymph nodes," "reactive adenopathy."
8.  "Emphysema" Definition:  Destruction of lung tissue leading to abnormally enlarged airspaces.  Example Clues:  "emphysematous lung changes," "bullous emphysema."
9.  "Atelectasis" Definition:  Collapse or incomplete expansion of lung tissue.  Example Clues:  "atelectasis" or "opacity likely representing atelectasis."
10.  "Lung nodule" Definition:  A small, round lesion within the lung parenchyma.  Example Clues:  Descriptions of "lung nodule" or specific measurements of nodules.

```
11.  "Lung opacity" Definition:  Areas of increased lung density (e.g.,
ground-glass opacities, consolidations, patchy opacities).  Example
Clues:  "ground-glass opacity," "consolidation," "patchy opacities."
12.  "Pulmonary fibrotic sequela" Definition:  Evidence of fibrotic
scarring in the lungs, such as reticulations or honeycombing.  Example
Clues:  "fibrotic lung disease," "honeycombing," "UIP pattern."
13.  "Pleural effusion" Definition:  Accumulation of fluid within the
pleural space.  Example Clues:  "pleural effusion" (specify if small,
moderate, or loculated).
14.  "Mosaic attenuation pattern" Definition:  Patchy areas of differing
lung attenuation that can indicate small airway disease.  Example Clues:
"mosaic attenuation" or "air trapping."
15.  "Peribronchial thickening" Definition:  Thickening of the tissues
surrounding the bronchi, often reflecting inflammation.  Example Clues:
"peribronchial wall thickening."
16.  "Consolidation" Definition:  Solidification of lung tissue due
to alveolar filling (by fluid, pus, blood, or cells).  Example Clues:
"consolidation" clearly stated or "no consolidation" when absent.
17.  "Bronchiectasis" Definition:  Permanent dilation of the
bronchial airways.  Example Clues:  "bronchiectasis" (e.g., "traction
bronchiectasis" or "varicoid bronchiectasis").
18.  "Interlobular septal thickening" Definition:  Thickening of the
septa between lung lobules.  Example Clues:  Phrases like "septal
thickening" or "interlobular septal thickening."
Instructions for Analysis:  - For each abnormality:  - If the report
explicitly describes or implies the abnormality, assign a value of 1.  -
If the report explicitly states that the abnormality is absent, assign
a value of 0.  - If the abnormality is not mentioned at all, assign a
value of -1.  - Use all details in the FINDINGS and IMPRESSION sections
to guide your labeling.  - Ensure the output JSON object contains all 18
keys exactly as listed.
Now, analyze the provided CT chest report and output a JSON object with
the 18 abnormality keys set to 1, 0, or -1 accordingly.  """
```

