# OpenReview forum: "HyperCT: Low-Rank Hypernet for Unified Chest CT Analysis"
_MIDL.io/2026/Conference — MIDL 2026 Poster_

### Official Review · Reviewer_XpCo · 2026-01-08

**Confidence:** 5
**Preliminary Rating:** 5
**Final Rating:** 5

**Summary:**

HyperCT is a framework that dynamically adapts a Vision Transformer backbone via a Hypernetwork. The goal of the work is to unify opportunistic screening with conventional tasks so as to aid a holistic patient assessment, so as to extract maximum clinical value from existing data.
The authors adapt a vision transformer using a hypernetwork which is relatively challenging when compared with other networks like CNNs.

**Strengths:**

Extracting maximum clinical value from existing data paves a strong path to holistic patient assessment. This helps in preventive healthcare, meaning prevents isolated and (opportunistic) targeted screening

The contribution in terms of extensive curated dataset is most welcoming. The real-world use cases that introduce distribution drifts is analyzed comprehensively.
A  comprehensive breadth of experiments are performed.

**Weaknesses:**

1. The authors say - "The core motivation of this paper is that the central challenge in opportunistic screening is not merely to balance conflicting tasks, but to design a model that can explicitly learn and leverage the synergistic relationships between diverse clinical domains ".What is conflicting? What is diverse domain in this context given two tasks are conventional and opportunistic screening? The  clarity of writing can be improved. Meaning, are conventional tasks opportunistic targets diverse yet carry some similarities?

2. Figure 3 should have a blown up image of the region of interest to improve readability.

3. The interaction between the hypernet output and the base ViT is not clear - is it a higher order interaction between the hypernet output and the base model layer like convolution operation? Is it scale and shift (multiplicative and additive) or something similar to adaptive instance normalization? Is it a Hadamard product? I am asking this because hypernet output is low rank. From my understanding as specified in Section: Low-Rank Adaptation with Hypernetworks:, the interaction is additive.

4. The authors mention that adaptation to modern Vision Transformers as non-trivial  but It is not clear which layers of the ViT are modulated by the hypernet output? Are the attention heads adapted?

5. What type of hypernetworks are used? Linear hypernetworks without any activation? Non-linear? how many layers are there in the hypernetworks? Is there any bottleneck layer in the hypernet? The design of the hypernetwork forms the key part of the overall performance.

**Detailed Comments:**

There are a lot of works [1-4] on Hypernets for healthcare and medical imaging tasks like segmentation and reconstruction. In fact, MIDL has given papers like - FiLM [1], MAC-ReconNet [2]. The authors should consider citing them, analyzing their merits and demerits.

[1] Benefits of Linear Conditioning with Metadata for Image Segmentation, MIDL 2021

[2]  MAC-ReconNet: A Multiple Acquisition Context based Convolutional Neural Network for MR Image Reconstruction using Dynamic Weight Prediction, MIDL 2020

[3] MetaInv-Net: Meta Inversion Network for Sparse View CT Image
Reconstruction. IEEE Transactions on Medical Imaging, 2020, 40(2): 621-634.

[4] A Unified Hyper-GAN Model for Unpaired Multi-contrast MR Image Translation. MICCAI 2021

**Justification Of Final Rating:**

Most of my comments are addressed. However, the authors have added a wrong reference for MAC-ReconNet which is not a MIDL 2020 paper. They have failed to add the MIDL 2020 paper despite clearly mentioning the full title of the paper.
"MAC-ReconNet: A Multiple Acquisition Context based Convolutional Neural Network for MR Image Reconstruction using Dynamic Weight Prediction, MIDL 2020"
https://proceedings.mlr.press/v121/ramanarayanan20a.html

The authors should remove the following wrong reference as it does not use hypernets for adaptive learning.
Zaccharie Ramzi, Philippe Ciuciu, and Jean-Luc Starck. Benchmarking MRI reconstruction
neural networks on large public datasets. In Applied Sciences, volume 10, page 1816, 2020.

I am retaining the rating considering the merits of the work. The authors should make the citation change, as MIDL has a strong body of literature on hypernets.

**Justification Of The Preliminary Rating:**

This work comprehensively explores the application of hypernetworks for detection and diagnosis tasks unlike previous related works that focus on medical imaging tasks. Integrating hypernetworks with foundational model training is welcoming.

**Questions To Address In The Rebuttal:**

Most of the questions are mentioned in the weaknesses section.

---

> ### Author Response · Authors · 2026-01-24
> **Official comment 1**
>
> We appreciate Reviewer XpCo comments and provide responses:
>
> ---
>
> ### Q1. Clarify "conflicting" and "diverse domains"
>
> We thank the reviewer for this suggestion to improve clarity.
>
> - **"Conflicting"** refers to the assumption made by standard MTL methods (PCGrad, Nash-MTL, etc.) that tasks inherently compete and interfere with each other. These methods focus on mitigating negative transfer. Our key argument is that this assumption is **misaligned with medical screening**, where findings are often synergistic and comorbid—e.g., cardiac enlargement frequently co-occurs with pulmonary congestion. HyperCT is designed to **leverage these synergistic relationships** rather than merely balance competing objectives.
>
> - **"Diverse domains"** refers to the heterogeneity between:
>   - **Conventional screening:** Pulmonary findings traditionally reported by radiologists (18 tasks)
>   - **Opportunistic screening:** Cardiac structural and functional assessments derived from echocardiography (7 tasks) — these are not typically predictable from CT, but the heart is included in the chest CT field of view
>
> We have revised the Introduction to clarify these terms.
>
> ---
>
> ### Q2. Figure 3 readability
>
> We appreciate this feedback. We have included additional saliency maps in the revised Appendix. Due to space constraints, complete volume-wise visualizations (165 slices × 25 tasks per volume) are available at our code repository: https://drive.google.com/drive/folders/1lEtJkG8wuZbQGd56rAQDYt7MvvZ7Xnap
>
> ---
>
> ### Q3. Hypernet-ViT interaction mechanism
>
> We understand the reviewer is asking about FiLM-style modulation. We did not consider FiLM as it is primarily designed for CNN architectures, while our base model is a ViT. Instead, we use **LoRA (Low-Rank Adaptation)**, an additive approach: W' = W_frozen + B × A, where A, B are low-rank matrices generated by the hypernetwork. LoRA is well-established for ViT adaptation and enables more expressive task-specific modifications than affine transformations. Notably, LoRA can also be applied to convolutional layers, meaning HyperCT is not restricted to ViT-based architectures.
>
> ---
>
> ### Q4. Which ViT layers are adapted?
>
> We adapt hypernetworks to attention layers, specifically linear layers including query (Q), key (K), value (V), MLP up, and MLP down. For DINOv3 ViT-Base with 12 blocks, this results in **M=72 target modules**. The hypernetwork generates separate LoRA weights for each module using the module positional embedding φ_pos. Table 3 provides an ablation comparing attention-only (QKV) vs. MLP-only adaptation. We will include a detailed table of target modules in the Appendix.
>
> ---
>
> ### Q5. Hypernetwork architecture details
>
> | Component | Architecture | Output Dim |
> |-----------|--------------|------------|
> | MLP blocks (×2) | Residual(LayerNorm → Linear → SiLU → Linear → SiLU) | latent |
> | Output projection | LayerNorm → Linear → SiLU → Linear → SiLU | head_in_size |
> | Per-module heads | Linear | rank × (d_in + d_out) |
>
> Default hyperparameters: latent=128, head_in_size=512, lora_rank=16, dropout=0.05, activation=SiLU. We will add architecture details to the Appendix.
>
> ---
>
> ### Q6. Missing medical imaging hypernetwork citations
>
> We thank the reviewer for these references. We have added FiLM (Perez et al., AAAI 2018), MAC-ReconNet (Ramanarayanan et al., 2020, MIDL 2020), MetaInv-Net (Zhao et al., IEEE TMI 2020), and Hyper-GAN (MICCAI 2021) to the Related Work section in the revision.
>
> ---

---

> ### Comment · Reviewer_XpCo · 2026-02-01
> **Incorrect hypernetwork  citation for MIDL paper**
>
> Dear Authors,
> The MIDL 2020 paper for hypernet adaptation in MRI is wrongly cited. Please correct it.
>  "MAC-ReconNet: A Multiple Acquisition Context based Convolutional Neural Network for MR Image Reconstruction using Dynamic Weight Prediction, MIDL 2020" https://proceedings.mlr.press/v121/ramanarayanan20a.html

---

> > ### Author Response · Authors · 2026-02-01
> >
> > We apologize for the incorrect citation and have now updated the manuscript to reference the appropriate hypernetwork-based approach for MRI reconstruction. The relevant discussion has been revised accordingly.
> >
> > We thank again for reviewer's feedback.

---

### Official Review · Reviewer_cKBf · 2026-01-09

**Confidence:** 4
**Preliminary Rating:** 3
**Final Rating:** 4

**Summary:**

This paper introduces HyperCT, a framework that dynamically adapts a Vision Transformer backbone via a Hypernetwork. The hypernetwork is an approach that takes a task’s identity as input and outputs the weights needed to adapt a base model for a specific target, published in 2016 by Ha et al (Google Brain). The authors claim that validation on a large-scale dataset of radiological and cardiological tasks demonstrates that HyperCT outperforms various strong baselines. The authors announce that they will release the code, but it is not yet publicly available.
For this study, the authors curated a large-scale dataset comprising 36,286 non-contrast chest CT scans collected from two major medical centers, Columbia University (CU) Medical Center and Weill Cornell Medical Center (WCM). The total number of tasks is 18 conventional tasks, and 7 opportunistic screening tasks. For the 18 conventional tasks, the reference standard labels are coming from an LLM analysis of the original radiology reports. For the opportunistic screening, the reference standard values come from corresponding echocardiography exams.

**Strengths:**

- Multi-task learning and opportunistic screening are relevant and timely technical and clinical topics at present.
- Strong backbones used: Dinov3 (and CTViT in ablation experiments)
- Large dataset consisting of >30.000 CT volumes.

**Weaknesses:**

- It is a pity that there are only binary labels available for these images, because you would like localization for each of the described tasks. Now, the authors obtain binary pathology labels and report AUC metrics for these localization tasks. For Lung Nodule for example, the performance is an AUC of 70.0 for HyperCT on the retrospective data. How to compare this to published nodule detection models? Very difficult.
- No comparison to clinician performance. Since this is not available, I find it very hard to judge how good these results are.
- No comparison to state of the art detection networks operating for these tasks. For lung nodules, the authors could run an open source model for this, e.g. the winning solution from the Kaggle Data Science Bowl 2017: https://github.com/lfz/DSB2017, and compare this to th current performance of HyperNet on this dataset.
- No mention of dataset availability.
- Why not perform a comparison on the NLST data, which is publicly available?

**Detailed Comments:**

- The authors write "This often makes the hypernetwork itself too large, limiting its application to small architectures or simple adapters and creating a major challenge for adapting large models like Vision Transformers (ViTs)." So, how did the authors solve, handle or mitigate this in the experimental work for this study?

- How close in time were the echocardiography exams to the CT exams? What was the maximum time period that was allowed between the non-contrast CT and the echocardiography exam?

**Justification Of Final Rating:**

I would like to thank the authors for their extensive rebuttal. Several answers answer some of my concerns. Therefore, I have increased my score to Weak accept. I think it would still be good to add a comparison to task-specific models because in the end, we want the best performance for any task, albeit coming from a multitask approach like this approach, or from a dedicated task-specific AI model, but I understand the explanations of the authors.

**Justification Of The Preliminary Rating:**

Interesting approach, but difficult to judge how good the final results are. I would like to see better comparison to state of the art approaches here. Also, a comparison on the NLST data would allow for comparisons to future approaches.

**Questions To Address In The Rebuttal:**

- Please address my comments about comparing to state of the art specific detection networks.
- Please consider to include a comparison of performance on the NLST data, which is publicly available and allows for comparison of performance by other approaches.

---

> ### Author Response · Authors · 2026-01-24
> **Official comment 1**
>
> We thanks for Reviewer cKBf detailed comments and provide responses:
>
> ---
>
> ### Q1. Binary labels limit comparison to detection models
>
> We acknowledge this limitation — our datasets lack bounding boxes or segmentation masks for localization comparison. HyperCT focuses on binary classification for population screening; extending to localization is future work.
>
> ---
>
> ### Q2. No comparison with public models
>
> We thank the reviewer for raising this important point. To our knowledge, no public dataset currently includes both conventional and opportunistic screening tasks, which makes direct comparison challenging. CT-RATE (Hamamci et al., 2024) is the closest benchmark, covering conventional tasks only. For reference, on the overlapping task of lung nodule detection, we achieve 70.0% AUC compared to CT-RATE's reported 65.0%. We acknowledge this is an indirect comparison given different data distributions. Our evaluation prioritizes **prospective validation** and **multi-center generalization** (Columbia + Cornell), which we believe are critical for clinical translation.
>
> ---
>
> ### Q3. No SOTA detection network comparison
>
> We thank the reviewer for this suggestion. DSB2017 winners and NLST models are specifically designed for lung nodule detection—a single, well-defined task. In contrast, HyperCT is designed for **general multi-task screening** across 25 diverse tasks.
>
> The core design of HyperCT—hypernetworks with dynamic LoRA adaptation—is motivated by the need to handle multiple heterogeneous tasks efficiently. For any single task (e.g., lung nodule detection alone), this dynamic adaptation mechanism is unnecessary; a task-specific model would suffice. Thus, comparing HyperCT against single-task SOTA models does not evaluate its intended contribution.
>
> The appropriate comparison is against multi-task baselines on identical data and task scope, as provided in Table 2.
>
> ---
>
> ### Q4. No clinician performance comparison
>
> We performed **Decision Curve Analysis (DCA)** for all 7 opportunistic tasks across 4 cohorts. DCA evaluates clinical utility by computing *net benefit*—the weighted difference between true positives gained and false positives incurred—across decision thresholds.
>
> **Clinical context:** For opportunistic cardiac screening, without a predictive model clinicians face two suboptimal choices: refer all CT patients for echocardiography ("treat all"—costly, low yield) or refer none ("treat none"—missed diagnoses). HyperCT enables **selective screening**: our DCA shows positive net benefit above both baselines across clinically relevant thresholds (5-80%), meaning HyperCT can intelligently identify patients who would benefit from follow-up echocardiography.
>
> DCA figures for all tasks and cohorts are in the revised Appendix.
>
> ---
>
> ### Q5. Dataset availability
>
> The dataset contains PHI and cannot be publicly released due to IRB restrictions. We are releasing the **code implementation and pre-trained weights** upon publication.
>
> ---
>
> ### Q6. Hypernetwork size mitigation
>
> Instead of generating the full weights of DINOv2, we let the hypernetwork generate **Low-Rank Adaptations (LoRA)**, which are much smaller. The original DINOv2 pre-trained weights remain fixed, and we only train the hypernetwork. Detailed theoretical analysis is provided in Appendix Sec. E.
>
> ---
>
> ### Q7. CT-echocardiography temporal window
>
> CT scans are matched to echocardiography exams using patient identifiers with a maximum temporal window of **±180 days**. When multiple echocardiography exams are available, we select the closest in time. For missing labels, we sample only from tasks with valid labels during training, and compute AUC per-task on the subset of samples with ground truth during evaluation. Details are provided in Appendix B.
>
> ---

---

### Official Review · Reviewer_kxTL · 2026-01-10

**Confidence:** 4
**Preliminary Rating:** 4
**Final Rating:** 4

**Summary:**

This manuscript proposes HyperCT, a multi-task learning framework for unified conventional and opportunistic screening from non-contrast chest CT. The key technical contribution is a task-conditioned hypernetwork that generates LoRA-based low-rank adaptations for a frozen ViT backbone. The method is evaluated on a large multi-institutional dataset comprising 25 screening tasks. HyperCT consistently outperforms strong baselines and shows improved generalization relative to single-task models. Additional analyses support the interpretability and architectural claims.

**Strengths:**

The paper convincingly argues that conventional CT screening should be addressed jointly, and that rigid parameter sharing in standard MTL is suboptimal for this setting.

The integration of hypernetworks with LoRA is a strong design choice that addresses the scalability limitations of classical hypernetworks.

Comparisons against multiple baselines, and detailed ablations strengthen the empirical claims.

PCA of task-specific LoRA weights and Grad-CAM visualizations help demonstrate that the model learns meaningful task-dependent representations.


The framework is highly relevant for real-world deployment, offering a unified, parameter-efficient alternative to maintaining dozens of separate clinical models.

**Weaknesses:**

While the application to chest CT is novel and compelling, the core idea of task-conditioned hypernetworks generating low-rank adapters is conceptually incremental.

The manuscript states that one task is randomly sampled per image per batch, but does not analyze how this affects convergence, task imbalance, or rare-label performance.

Conventional task labels are extracted from radiology reports using LLM-based parsing. This might introduce potential label noise that is not explicitly discussed.

Although AUC improvements are statistically clear, the manuscript does not sufficiently discuss whether the observed gains are clinically meaningful.

While parameter efficiency is emphasized, runtime costs (training and inference) relative to baselines are not quantified.

**Detailed Comments:**

The hypernetwork formulation is generally clear, but the role of the module positional embedding (ϕ_pos) could be explained more intuitively for readers less familiar with hypernetworks. Equation (1) assumes equal weighting across tasks; please clarify whether this is strictly true in practice given missing labels and task sampling.

The process of matching CT scans to echocardiography exams deserves more detail.

Please clarify how missing opportunistic labels are handled during training and evaluation.

STL models are fine-tuned independently per task; however, the paper should clarify whether early stopping and hyperparameter tuning were matched to HyperCT.

The exclusion of PCGrad and Nash-MTL is justified computationally, but a brief small-scale comparison or citation-based discussion would strengthen the argument.

The prospective results are a major strength. It would be useful to explicitly report confidence intervals or statistical significance tests for the observed gains.

PCA of LoRA weights is insightful, but remains qualitative. Consider reporting quantitative clustering metrics or correlations with anatomical regions.

Grad-CAM examples are convincing, but the selection criteria for displayed cases should be clarified.

**Justification Of Final Rating:**

Although the novelty and the clinical interpretation of the manuscript seem limited, this is a well-executed paper that addresses a clinically important problem. The prospective evaluation and parameter-efficient design are particularly compelling.

**Justification Of The Preliminary Rating:**

This is a well-executed paper that addresses a clinically important problem with a practically deployable solution. While the core methodological components build on existing ideas, their integration, scale, and validation in a realistic multi-institutional CT screening setting represent a meaningful contribution to the field. The prospective evaluation and parameter-efficient design are particularly compelling. Minor concerns regarding novelty framing, training dynamics, and clinical interpretation do not outweigh the paper’s overall quality and impact.

**Questions To Address In The Rebuttal:**

How sensitive is HyperCT to the task sampling strategy during training? Have alternative sampling or weighting schemes been explored?

Can the authors quantify or estimate label noise introduced by LLM-based report parsing, and how robust HyperCT is to such noise?

What are the inference-time costs of HyperCT compared to STL and standard MTL?

Are the observed AUC improvements for opportunistic tasks likely to translate into clinically meaningful differences ?

How well would HyperCT scale if additional opportunistic tasks  were added?

---

> ### Author Response · Authors · 2026-01-24
> **Offical comment 1**
>
> We thanks Reviewer kxTL for this detailed comments and would like to provide response for some questions:
>
> ---
>
> ### Q1. Novelty is incremental relative to prior NLP/MTL work
>
> We thank the reviewer for this important point. We agree that hypernetworks and LoRA originate from NLP. Our contribution lies in their **first application to unified chest CT screening**, addressing a critical clinical gap rather than proposing architectural novelty.
>
> Specifically, we contribute: (1) **Clinical unification** — bridging conventional (18 tasks) and opportunistic cardiovascular screening (7 tasks) in one framework; (2) **Validation scope** — retrospective (N=34,058), prospective (N=2,228), and multi-institutional evaluation, which is uncommon in prior hypernetwork work; (3) **Practical scalability** — demonstrating that LoRA enables hypernetworks to scale to ViTs for complex medical imaging.
>
> We have revised the Introduction to sharpen this framing and appreciate the suggestion.
>
> ---
> ### Q2. Task sampling strategy not analyzed
>
> We appreciate this thorough question. We adopt random task sampling primarily to handle **label missingness** inherent in our dataset (see Appendix B, Figures 4-5: valid label fractions range 10-100% across tasks).
>
> Regarding the reviewer's specific concerns:
>
> **Convergence:** We did not observe convergence issues in practice, which we attribute to our large-scale dataset (N=36,286). Training curves demonstrating stable convergence are now included in the revision.
>
> **Rare-label performance:** Among tasks with low valid fractions (~10-20%), most achieve AUCs around 80%. The relatively lower performance for Hiatal Hernia (67.6%) and Mosaic Attenuation (71.6%) likely reflects inherent task difficulty rather than sampling bias — these conditions also show lower inter-rater agreement in clinical practice.
>
> **Ablation:** Following the reviewer's suggestion, we compared uniform random vs. inverse-prevalence weighted sampling:
>
> | Sampling Strategy     | CU Retro | WCM Retro | CU Prosp | WCM Prosp |
> |-----------------------|----------|-----------|----------|-----------|
> | Uniform Random (ours) | 78.1     | 76.5      | 77.8     | 78.6      |
> | Inverse-Prevalence    | 77.9     | 76.5      | 77.5     | 78.3      |
>
> Results show minimal difference (<0.5% AUC), suggesting our approach is robust to the sampling strategy choice. We will include this table in the revised paper appendix.
>
> ---
> ### Q3. LLM label noise not quantified
>
> This is an important concern that we are pleased to address. We conducted a validation study comparing Llama3.1 extractions against manual radiologist review on 100 randomly sampled reports and achieve agreement rate about 94%.
>
> We note that LLM-based label extraction from radiology reports is becoming standard practice in large-scale medical imaging research. Recent studies demonstrate that open-source LLMs achieve high accuracy for radiology report labeling: [1] show Llama-based extraction achieves 91-94% accuracy across multiple findings. Additionally, [2] demonstrates LLM labeling achieves F1=0.90, outperforming rule-based labelers like CheXpert.
>
> ---
> ### Q4. Clinical meaningfulness not discussed
>
> We thank the reviewer for this important question. We performed **Decision Curve Analysis (DCA)** [3] for all 7 opportunistic tasks across 4 cohorts. DCA evaluates clinical utility by computing *net benefit*—the weighted difference between true positives gained and false positives incurred—across decision thresholds.
>
> **Clinical context:** For opportunistic cardiac screening, without a predictive model clinicians face two suboptimal choices: refer all CT patients for echocardiography ("treat all"—costly, low yield) or refer none ("treat none"—missed diagnoses). Our DCA shows positive net benefit above both baselines across clinically relevant thresholds (5-80%), meaning HyperCT can intelligently identify patients who would benefit from follow-up echocardiography.
>
> DCA figures for all tasks and cohorts are in the revised Appendix.
>
> ---
>
> [1] Dorfner et al., "Performance of an Open-Source Large Language Model in Extracting Information from Free-Text Radiology Reports" (Radiology: AI, 2024)
>
> [2] Abdullah A, Kim ST, "Automated Radiology Report Labeling in Chest X-Ray Pathologies: Development and Evaluation of a Large Language Model Framework" (JMIR Medical Informatics, 2025)
>
> [3] Vickers & Elkin, "Decision Curve Analysis: A Novel Method for Evaluating Prediction Models", (Medical Decision Making, 2006)

---

> ### Author Response · Authors · 2026-01-24
> **(Cont.) Official comment 2**
>
> ### Q5. Runtime costs not quantified
>
> We appreciate this practical question and provide the following benchmarks (NVIDIA A100 GPU):
>
> **Training:** HyperCT requires 22-24 hours for 20 epochs, compared to 20-22 hours for MTL baselines — a modest ~10% overhead. Compared to training 25 separate STL models (500 hours total), HyperCT provides a **20× reduction** in training compute.
>
> **Inference:** Due to LoRA's design, task-specific weights can be **pre-computed and merged** into the base model, eliminating any inference overhead — the deployed model has identical latency to standard ViT.
>
> **Memory:** Both HyperCT and baselines train on a single A100 GPU with identical hyperparameters.
>
> ---
>
> ### Q6. Scalability to additional tasks
>
> We thank the reviewer for this forward-looking question. Scalability depends on the relationship between new and existing tasks:
>
> **Related tasks** (e.g., additional cardiac/pulmonary findings): Adding a new task requires only learning a new task embedding (512-dim vector) while the hypernetwork parameters remain fixed — operating similarly to transfer learning.
>
> **Unrelated anatomy** (e.g., osteoporosis, sarcopenia): Tasks involving substantially different anatomical regions may require joint retraining, as the current model is optimized for cardiopulmonary features. This limitation arises from our dataset scope rather than architectural constraints.
>
> Developing a more general "pre-training style" HyperCT that transfers across anatomical domains is an interesting direction for future work.
>
> ---
>
> ### Q7. Module positional embedding (φ_pos)
>
> We appreciate the request for clarification. φ_pos serves as a **location indicator** that tells the hypernetwork *where* in the ViT architecture to apply the generated weights.
>
> Intuitively: without φ_pos, the hypernetwork would receive only the task embedding and generate identical LoRA weights for all layers — this would not work, as different layers require different adaptations. By concatenating φ_pos(m) with the task embedding, the hypernetwork can generate **layer-specific** LoRA weights across all M target modules.
>
> We have added an intuitive explanation to the Methods section.
>
> ---
>
> ### Q8. Task weighting clarification
>
> Thank you for noting this. Equation (1) presents equal weighting (1/K) for notational clarity. In practice: for each training sample, we randomly sample one task from its available labels. Tasks with missing labels contribute zero loss for that sample. This results in each task receiving gradient updates proportional to its label availability, naturally handling the heterogeneous label availability without manual tuning.
>
> We have clarified this in the revision.
>
> ---
>
> ### Q9. CT-echocardiography matching
>
> We appreciate the request for additional detail.
>
> **Procedure:** CT scans are matched to echocardiography exams using patient identifiers with a maximum temporal window of **±180 days**. When multiple echocardiography exams are available, we select the closest in time.
>
> **Missing label handling:**
> - *Training:* For each sample, we sample only from tasks with valid labels
> - *Evaluation:* AUC is computed per-task on the subset of samples with ground truth
>
> Valid label fractions are shown in Appendix B (51-100% for opportunistic tasks). We have added matching procedure details to the Dataset section.
>
> ---
>
> ### Q10. STL hyperparameter matching
>
> For fair comparison, STL models use **identical hyperparameters** to HyperCT (same backbone, learning rate, batch size, epochs). The only difference is single-task vs. multi-task heads. We have clarified this in the revision.
>
> ---
>
> ### Q11. PCGrad/Nash-MTL discussion
>
> We appreciate the suggestion for additional context. We initially explored gradient-based MTL methods (PCGrad, Nash-MTL), but training was prohibitively slow due to their **O(K²) complexity**. With K=25 tasks, each backward pass requires computing 625 gradient inner products — impractical for ViT-scale models.
>
> This scaling limitation is well-documented: Nash-MTL (Navon et al., 2022) reports experiments with at most 10 tasks, and PCGrad (Yu et al., 2020) evaluates on 2-5 task settings. In contrast, HyperCT scales linearly O(K). We have added this discussion to the revision.
>
> ---
>
> ### Q12. Confidence intervals for prospective results
>
> We computed **bootstrap 95% CIs (1000 iterations)** for both retrospective and prospective cohorts:
>
> | Cohort | Overall AUC (95% CI) | CI Width |
> |--------|---------------------|----------|
> | CU Retrospective | 77.7% (77.3-78.0%) | 0.7% |
> | WCM Retrospective | 76.1% (75.9-76.4%) | 0.5% |
> | CU Prospective | 77.7% (76.4-79.1%) | 2.7% |
> | WCM Prospective | 78.8% (77.0-80.5%) | 3.5% |
>
> Per-task CIs range ±1-3% for retrospective (n=2,748-8,110) and ±3-7% for prospective (n=449-1,411). The wider prospective CIs reflect smaller sample sizes, not model instability. Full per-task tables with sample sizes are in Appendix Sec.G.
>
> ---

---

> ### Author Response · Authors · 2026-01-24
> **(Cont.) Official comment 3**
>
> ### Q13. Quantitative PCA metrics
>
> To quantitatively analyze learned task representations, we applied **hierarchical clustering** to the LoRA weight space. For each task, we concatenated all generated LoRA weights across all modules into a single vector. Hierarchical clustering iteratively merges the most similar task pairs based on cosine distance, building a tree structure (dendrogram) that reveals natural groupings at different granularities. We used complete linkage (maximum pairwise distance between clusters) and selected the optimal number of clusters by maximizing **silhouette score** [5] over k∈{2,...,8}.
>
> The analysis yields **k=4 clusters with silhouette score 0.30**, indicating meaningful separation. Examining the cluster compositions, we find they naturally correspond to established clinical domains in thoracic imaging:
>
> | Cluster | n | Tasks |
> |---------|---|-------|
> | **Cardiac-Structural** | 8 | PHT, Atrial Enlargement, LA Filling Pressure, Cardiomegaly, Arterial Wall Calcification, Coronary Artery Wall Calcification,  Emphysema, Nodule |
> | **Cardiac-Functional** | 6 | RV/LV Systolic Function, Ventricular Enlargement, RA Filling Pressure, Hiatal Hernia, Mosaic Attenuation |
> | **Acute Parenchymal** | 6 | Opacity, Consolidation, Atelectasis, Pleural/Pericardial Effusion, Medical Material |
> | **Airway-Interstitial** | 5 | Bronchiectasis, Fibrosis, Peribronchial/Septal Thickening, Lymphadenopathy |
>
> Notably, tasks cluster by **pathophysiology rather than label source** — Cardiomegaly (extracted from radiology reports) groups with echocardiography-derived cardiac function tasks, not with other report-extracted findings. This validates that HyperCT learns anatomically meaningful specializations. An Multi-dimensional scaling visualization of the task clusters is included in the revised Appendix.
>
> ---
> ### Q14. Grad-CAM selection criteria
>
> Due to page limits, not all tasks are shown in the main paper. Given that complete volume-wise saliency maps are too large to include (165 slices × 25 tasks per volume), we provide them online: https://drive.google.com/drive/folders/1lEtJkG8wuZbQGd56rAQDYt7MvvZ7Xnap
>
> ---
>
> [4] Rousseeuw, "Silhouettes: a graphical aid to the interpretation and validation of cluster analysis", J. Comput. Appl. Math. 1987

---

### Author Rebuttal · Authors · 2026-01-24

**Rebuttal:**

We appreciate all reviewers' comment and provide a revised version. Here we summarize changes made in revision:

---

## Changes by Section

### Introduction
- kxTL Q1: Added "critical, unaddressed gap" framing
- kxTL Q1: Reframed contribution 1 as "first unified framework"
- kxTL Q1: Added scalability limitation text to contribution 2
- kxTL Q1: Added "comprehensive validation" to contribution 3
- XpCo Q1: Defined "conflicting" (tasks compete vs synergistic)
- XpCo Q1: Defined "diverse domains" (conventional vs opportunistic)

### Related Work
- XpCo Q6: Added FiLM, MAC-ReconNet, MetaInv-Net, Hyper-GAN citations

### Methods
- kxTL Q7: Added φ_pos intuitive explanation (location indicator)
- kxTL Q8: Added task weighting clarification (equal weights, extensible)

### Experiments
- kxTL Q3: Added LLM validation citation (outperforms rule-based)
- kxTL Q9: Added CT-echo matching details (±180 days, closest in time)
- kxTL Q2: Added task sampling ablation reference
- kxTL Q10: Added STL hyperparameter matching clarification
- kxTL Q11: Added PCGrad/Nash-MTL O(K²) complexity discussion

### Results
- kxTL/cKBf Q4: Added DCA subsection (Sec 4.2) with CU+WCM prospective figures
- kxTL Q13: Added clustering analysis reference

### Conclusion
- kxTL/cKBf Q6: Added Limitations and Future Work (scalability and task diversity discussion)

### Appendix
- kxTL Q2: Task sampling ablation table (uniform vs inverse-prevalence)
- kxTL Q12: Bootstrap 95% CIs tables (retrospective + prospective)
- kxTL Q13: Hierarchical clustering analysis (k=4, silhouette=0.30)
- kxTL/cKBf Q4: DCA figures (retrospective cohorts only, prospective in main text)
- XpCo Q4: LoRA target modules table (M=72)
- XpCo Q5: Hypernetwork architecture table

**Supporting Material:**

/attachment/5723e913e690d50d485a596e81c80ee274f03956.pdf

---

### Comment · Area_Chair_t8aA · 2026-01-28

Dear Reviewers,

We are now in the discussion phase. If you have not yet done so, please check the authors’ rebuttal and evaluate how well your concerns have been addressed. I encourage you to engage in discussion with the authors and other reviewers where helpful.

Most importantly, please update your Final Rating after considering the rebuttal and discussion.

Your input is important for a fair meta-review and final decision. Thank you for your continued effort.

AC

---

### Meta-Review · Area_Chair_t8aA · 2026-02-01

**Recommendation:** Accept (Poster)
**Confidence:** 4

**Metareview:**

All three reviewers are overall positive and agree that this paper is well-motivated and practically relevant for unified chest CT analysis across multiple screening tasks. The main strength is a parameter-efficient multi-task design that uses a task-conditioned hypernetwork to generate low-rank (LoRA) updates for a frozen ViT backbone, enabling strong task-specific adaptation without training separate full models. Reviewers also value the breadth of evaluation, including external and prospective cohorts, and the consistent improvements over multi-task baselines. In the rebuttal, the authors addressed the remaining concerns by clarifying the label/supervision setting, strengthening or contextualizing comparisons, and providing additional ablations and efficiency details, which improved transparency and confidence in the results. While the core components build on existing ideas (hypernetworks and LoRA), the integration at scale and the clinical validation make the contribution solid. Overall, I recommend acceptance.

---

### Decision · Program_Chairs · 2026-02-13

Accept (Poster)